# AcuRank: Uncertainty-Aware Adaptive Computation for Listwise Reranking

**Soyoung Yoon**[*]
Seoul National University
soyoung.yoon@snu.ac.kr

**Gyuwan Kim**[*]
University of California, Santa Barbara
gyuwankim@ucsb.edu

**Gyu-Hwung Cho**
Seoul National University
jgh0815@snu.ac.kr

**Seung-won Hwang**[†]
Seoul National University
seungwonh@snu.ac.kr

## Abstract

Listwise reranking with large language models (LLMs) enhances top-ranked results in retrieval-based applications. Due to the limit in context size and high inference cost of long context, reranking is typically performed over a fixed size of small subsets, with the final ranking aggregated from these partial results. This fixed computation disregards query difficulty and document distribution, leading to inefficiencies. We propose AcuRank, an adaptive reranking framework that dynamically adjusts both the amount and target of computation based on uncertainty estimates over document relevance. Using a Bayesian TrueSkill model, we iteratively refine relevance estimates until reaching sufficient confidence levels, and our explicit modeling of ranking uncertainty enables principled control over reranking behavior and avoids unnecessary updates to confident predictions. Results on the TREC-DL and BEIR benchmarks show that our method consistently achieves a superior accuracy–efficiency trade-off and scales better with compute than fixed-computation baselines. These results highlight the effectiveness and generalizability of our method across diverse retrieval tasks and LLM-based reranking models.[3]

## 1 Introduction

Modern information retrieval pipelines, such as web search and retrieval-augmented generation (RAG) systems, typically adopt a fast first-phase retriever, selecting a set of broadly relevant documents, like BM25 [1] or dense encoders [2] optimized for recall and speed. As they often produce noisy or suboptimal rankings, reranking is critical in applications where precision at the top is essential, including conversational agents and reasoning with large language models (LLMs) [3, 4].

Reranking methods can be broadly categorized by how they model document interactions and handle relative relevance. Pointwise methods [5] independently assign scores to each document, offering scalability but failing to consider the competitive context among candidates. Pairwise methods [6] improve upon this by comparing document pairs to capture local preferences, but often struggle to maintain coherent global rankings. Listwise methods [7, 8] instead evaluate a group of documents jointly, particularly effective for out-of-domain [9] or low-resource [10] settings. Recent work has shown that LLMs are especially effective as listwise rerankers, due to their ability to capture complex reasoning and fine-grained distinctions across documents [7, 11, 12].

---

[*]Equal contribution. Author order is randomly determined via coin toss.

[†]Corresponding author.

[3]https://github.com/soyoung97/AcuRank

39th Conference on Neural Information Processing Systems (NeurIPS 2025).

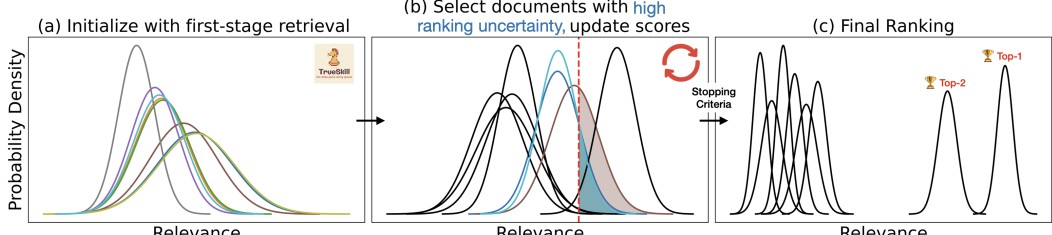

Figure 1: Overview of AcuRank. **(a)**: Each retrieved document's relevance is initialized as a Gaussian distribution using a TrueSkill-based model, with its mean and variance representing estimated relevance and uncertainty. **(b)**: We estimate the probability of a document being in the top-$k$ as the chance its score exceeds a threshold such that the expected number of documents above it equals $k$. Documents with uncertain rankings are selected and reranked in groups, and their relevance estimates are updated. **(c)**: The process repeats until stopping criteria are met. The final ranking is based on the updated relevance estimates.

Despite their effectiveness, applying LLMs in listwise reranking is computationally expensive. As each LLM reranker call can process only a small group of candidates due to input length limits, covering the full list typically requires multiple calls. Prior work has proposed fixed-computation strategies such as sliding windows [13] and tournament-style reranking [14] to balance ranking effectiveness with computational cost. While these fixed strategies improve efficiency, they suffer from rigidity and lack of adaptivity by assigning a fixed number of reranker calls to predetermined candidate positions, regardless of the complexity or ambiguity of the query. As a result, documents ranked low in early iterations may never be reconsidered, even when initial decisions are based on limited or noisy context. Moreover, these methods do not leverage intermediate signals from previous reranking steps and cannot dynamically focus computation on uncertain cases where refinement is most needed.

To address these limitations, we propose **AcuRank**, a framework for listwise reranking with **U**ncertainty-aware **A**daptive **C**omputation (rearranged for pronunciation). Our method adaptively allocates computation based on each document's relevance uncertainty relative to the top-$k$ boundary, rather than treating all documents equally. As illustrated in Figure 1, we adopt TrueSkill [15], a Bayesian rating system, to maintain probabilistic relevance estimates (Figure 1(a)) and iteratively update them as reranking results are observed. At each step, only uncertain candidates are selected for reranking (Figure 1(b)), while confident ones are skipped. This targeted refinement continues until convergence (Figure 1(c)), improving both accuracy and computational efficiency by focusing resources on the most ambiguous cases.

We evaluate our method on the TREC Deep Learning [16] and BEIR [17] benchmarks using multiple LLM-based rerankers. Our approach consistently improves ranking quality while requiring fewer reranker calls across diverse datasets and reranker models, including both in-domain and out-of-domain settings. These results demonstrate the effectiveness of uncertainty-aware adaptive computation and the broad applicability of our method.

Our contributions are as follows:

- We introduce probabilistic relevance modeling based on TrueSkill, which enables estimation of uncertainty in document rankings.
- Building on this, we develop a novel listwise reranking framework that supports adaptive computation by selectively reranking uncertain candidates through iterative refinements.
- Through extensive experiments, our method consistently improves the accuracy–efficiency trade-off across benchmarks and generalizes well across datasets and reranker models.

## 2 Related work

### 2.1 Fixed-computation reranking strategies

**Pointwise, pairwise, and setwise.**   Pointwise methods [18–21] score each document independently, treating reranking as a classification problem. They are efficient and compatible with neural encoders and LLMs but cannot model cross-document dependencies, limiting their effectiveness in complex

scenarios. Pairwise methods [22, 6, 23] compare two documents at a time to model pairwise dependency, but require $n^2$ number of reranker calls to rank $n$ documents. Setwise methods [24] are typically compared against pairwise methods, which rerank small fixed-size sets to balance between pointwise simplicity and listwise interaction. While efficient, they typically view only one or two passages at a time, limiting their ability to model subtle inter-document relationships visible only in larger listwise contexts. A concurrent OpenReview submission explores a *budget-constrained pairwise* reranking strategy that selects informative comparisons using TrueSkill-based uncertainty sampling; unlike our approach, it operates under a fixed pairwise budget rather than allocating computation adaptively over listwise contexts [25].

**Listwise - sliding windows.** Sliding window methods rerank overlapping subsets of candidates iteratively, starting from the bottom window of the list and progressively moving to the next window, merging top results upward. This approach has been widely adopted in LLM-based reranking systems such as LRL [13], RankGPT [11], RankVicuna [8], and RankZephyr [12], which use zero-shot prompts to perform reranking over windowed chunks. While effective, fixed window sizes and traversal paths lead to inefficiencies, by incurring over-computation for simple queries with little ambiguity and under-computation for complex queries.

**Listwise - tournament-style.** Tournament-style methods rerank by partitioning candidates into small groups and performing listwise reranking in iterative rounds. TDPart [26] employs a top-down partitioning strategy to minimize redundant inference and enhance parallel processing. In each round, listwise reranking is applied to the groups, and the top-ranked documents advance. This process repeats over progressively smaller pools until the final ranking is achieved. While TDPart focuses on efficiency, ListT5 [27] and TourRank [14] leverage T5 models and LLMs for groupwise comparisons, achieving competitive ranking accuracy. However, like sliding window methods, they are limited by fixed schedules, leading to over-computation for straightforward queries and under-computation for complex, ambiguous ones.

Complementary to scheduling-based approaches, CoRanking [28] integrates rerankers of different model sizes to reduce end-to-end cost, while DemoRank [29] explores demonstration selection tailored for ranking tasks. Both are orthogonal to our uncertainty-driven adaptive computation, yet potentially compatible with it.

## 2.2 Uncertainty and adaptivity in retrieval

Uncertainty estimation in retrieval has emerged as a principled method for enhancing reliability in retrieval. Cohen et al. [30] introduce a Bayesian framework that interprets relevance scores in deep retrieval models as stochastic estimates rather than deterministic values. Using techniques like dropout-based variational inference, they capture predictive uncertainty, leading to improvements in robustness, calibration, and resilience to adversarial inputs. Their findings expose the risks associated with overconfident ranking models and highlight the need for uncertainty-aware mechanisms in downstream tasks, such as filtering and reweighting. While post-hoc calibration over static retrieval outputs is allowed in their problem context, our work requires online computation for the iterative reranking setting. Our distinction lies in integrating the rigor of Bayesian models, inspired by TrueSkill [15], into adaptive computation, thereby making our approach both theoretically grounded and practically efficient. Meanwhile, prior adaptive computation methods [31] in retrieval are largely confined to statistical ranking scenarios [32] that depend on separate estimators for judging relevance thresholds or query difficulty [33–36].

In contrast, our method unifies uncertainty estimation and adaptive computation in a single framework: we use uncertainty not only as a measure of model confidence, but also as a dynamic signal that guides the allocation of computation. This enables fine-grained reranking that selectively refines ambiguous documents while avoiding unnecessary inference on confident ones.

## 3 Method

We begin by formalizing the reranking task (§3.1) and introducing our approach to relevance modeling and uncertainty quantification using TrueSkill (§3.2). Building on this foundation, we then describe the components of our reranking algorithm with adaptive computation, AcuRank (§3.3, Fig. 1).

While conventional reranking methods rely on a fixed number of reranker calls per query (§2.1), our method adaptively determines both which documents to rerank and how much computation to allocate, based on evolving document-level uncertainty estimates. This enables focused refinement of uncertain candidates, improving both computational efficiency and ranking effectiveness by avoiding unnecessary updates to already confidently ranked documents.

## 3.1 Problem formulation

Given a query $q$ and a set of retrieved documents $\mathcal{D} = \{D_1, \ldots, D_n\}$ from a first-stage retriever such as BM25, the goal is to reorder the documents into a ranked list $[D_{r_1} > \cdots > D_{r_n}]$ or to extract a top-$k$ list $[D_{r_1} > \cdots > D_{r_k}]$, where $\{r_1, \ldots, r_n\}$ is a permutation of $\{1, \ldots, n\}$.

We assume access to a listwise reranker $g(\mathcal{D}'; \mathcal{M})$, where $\mathcal{D}'$ is an ordered subset of $\mathcal{D}$ and $\mathcal{M}$ denotes the underlying reranking model. This is done by prompting an instruction-tuned or zero-shot LLM, and the reranker returns a new ordering over the documents in $\mathcal{D}'$.

In practice, $n$ is often large (e.g., 100–1000), making it infeasible to rerank all retrieved documents at once due to input sequence length constraints. Instead, $g$ is applied to smaller subsets with $|\mathcal{D}'| = m \ll n$, where $m$ is typically 20, and the final ranking is approximated by aggregating local reranking results across multiple batches. The ranked result is then evaluated using metrics such as NDCG@10, which only considers the top-$k$ documents in the ranking, so $k$ is often set to a small number like 10. In RAG systems, $k$ can be determined by the number of documents intended for inclusion in the generated context.

## 3.2 TrueSkill-based relevance modeling and ranking uncertainty quantification

At the core of our method is a probabilistic model of document relevance based on TrueSkill [15], a principled Bayesian ranking system originally developed for multiplayer games. It is known to produce well-calibrated estimates of both scores and uncertainty, and has been widely adopted in various applications where reliable uncertainty quantification is crucial [37–39]. Similar to prior works, we recast documents as players, and when one document is ranked above another, it is treated as having won that match. Specifically, we model the latent relevance of each document $D_i$ as a Gaussian variable $x_i \sim \mathcal{N}(\mu_i, \sigma_i^2 + \beta^2)$, where $\mu_i$ represents the estimated relevance, $\sigma_i$ captures epistemic uncertainty, and $\beta$ is a fixed global parameter representing observation noise. Here, epistemic uncertainty reflects the model's lack of confidence in its current estimate and is expected to decrease as more reranking evidence accumulates through iterative updates. This probabilistic view aligns with the intuition of Cohen et al. [30], who emphasize the importance of modeling uncertainty in relevance estimation for more effective ranking decisions.

Each document's relevance is thus represented not as a point estimate but as a belief distribution, capturing both its expected value and uncertainty. Modeling document relevance as a distribution enables principled reasoning under uncertainty and supports more informed reranking decisions. Unlike traditional methods that rely solely on deterministic rankings, our approach maintains document-level belief distributions and quantifies their relative strength. This formulation allows the computation of informative quantities such as the probability that a document appears in the top-$k$ and the entropy of its rank distribution, which serve as signals for uncertainty-aware computation. The model can also incorporate external signals, such as retrieval confidence or prior knowledge of relevancy signals, when available.

To estimate rank probabilities, we express the rank of $D_i$ as $r_i = 1 + \sum_{j \neq i} \mathbf{1}_{x_j > x_i}$, i.e., the number of documents more relevant than $D_i$ under their latent scores. Since each $x_i$ is Gaussian, the exact probability that $D_i$ has rank $r$ can be computed via:

$$\mathbb{P}(r_i = r) = \int_{-\infty}^{\infty} p_i(x) \cdot \sum_{\substack{S \subseteq \mathcal{D} \setminus \{D_i\} \\ |S| = r-1}} \left[ \prod_{D_j \in S} F_j(x) \prod_{D_l \in \mathcal{D} \setminus (S \cup \{D_i\})} (1 - F_l(x)) \right] dx$$

where $p_i$ is the probability density function of $x_i$ and $F_j$ is the the cumulative distribution function of $x_j$. The summation inside the integral can be evaluated using dynamic programming with $O(n^2)$ complexity by tracking the distribution over how many documents are ranked above $D_i$. However, it

becomes costly and numerically unstable as $n$ grows, due to the need to compute dense integrals and multiply small probabilities.

To improve scalability, we adopt a more efficient approximation. We define a threshold $t(r)$ such that the expected number of documents whose relevance exceeds $t(r)$ equals the target rank $r$, i.e., $\sum_i \mathbb{P}(x_i > t(r)) = r$. The value of $t(r)$ depends on each document's relevance distribution $(\mu_i, \sigma_i)$, and since $t(r)$ is monotonic to $r$, we efficiently compute $t(r)$ via binary search. We then approximate the cumulative rank probability as $\mathbb{P}(r_i \leq r) \approx \mathbb{P}(x_i > t(r))$, which reflects the chance that $x_i$ exceeds the estimated top-$r$ threshold.

### 3.3 AcuRank framework

Algorithm 1 outlines AcuRank, a listwise reranking method that performs adaptive computation guided by probabilistic relevance modeling and uncertainty estimation. At each iteration, AcuRank identifies documents with uncertain rankings and focuses reranking efforts on them, updating their relevance estimates based on listwise reranker outputs.

---

**Algorithm 1** AcuRank: Uncertainty-Aware Adaptive Computation for Listwise Reranking

---

**Input:** Query $q$, retrieved documents $\mathcal{D} = \{D_1, \ldots, D_n\}$, listwise reranker $g$, target rank cutoff $k$
**Output:** Ranked list $[D_{r_1} > \cdots > D_{r_n}]$, with top-$k$ used downstream
 1: Initialize TrueSkill-based relevance scores $(\mu_i, \sigma_i)$ for all $D_i \in \mathcal{D}$
 2: **repeat**
 3:   Select candidate documents $\mathcal{C} \subset \mathcal{D}$ with high ranking uncertainty
 4:   Partition $\mathcal{C}$ into ordered groups $\{\mathcal{B}_1, \ldots, \mathcal{B}_b\}$
 5:   Apply listwise reranker $g$ to each $\mathcal{B}_j$ and update TrueSkill scores accordingly
 6: **until** $|\mathcal{C}|$ is small, top-$k$ converges, or computational budget is exhausted

---

**Initialization (Line 1).** Each document's relevance distribution $(\mu_i, \sigma_i)$ is initialized using first-stage retrieval scores, with $\mu_i$ set to the raw score and $\sigma_i = \mu_i/3$. This provides a more informative prior than uniform defaults (Supplementary Section B.4). The observation noise parameter $\beta$ remains constant throughout. In ablation experiments, we also compare with the standard TrueSkill initialization ($\mu_i = 25$, $\sigma_i = 25/3$).

**Uncertainty-based selection (Line 3).** To identify uncertain documents, we compute $s_i = \mathbb{P}(x_i > t(k))$ for each document, where $t(k)$ is the expectation threshold ($r = k$) defined at Section 3.2 and $\sum_i s_i = k$ by construction. All documents have $s_i = k/n$ for standard TrueSkill initialization, but when using first-stage retrieval scores, $s_i$ values may deviate from uniformity based on score distribution. As reranking progresses, these values polarize toward 0 or 1. We define the uncertain candidate set as $\mathcal{C} = \{D_i \mid \epsilon < s_i < 1 - \epsilon\}$ using a tolerance hyperparameter $\epsilon$, selecting documents with intermediate $s_i$ values that are neither confidently in nor out of the top-$k$.

**Partitioning (Line 4).** When the number of uncertain documents exceeds the reranker's capacity $m$, we partition the candidate set $\mathcal{C}$ into equally sized groups. We first sort $\mathcal{C}$ in descending order of $\mu_i$, resulting in $[D_{\pi_1}, \ldots, D_{\pi_{|\mathcal{C}|}}]$, where $\mu_{\pi_1} \geq \cdots \geq \mu_{\pi_{|\mathcal{C}|}}$. Then, we partition $\mathcal{C}$ into disjoint ordered lists $\mathcal{B}_j = [D_{\pi_i} \mid i \in [(j-1) \cdot m, \min(j \cdot m, |\mathcal{C}|))]$, where $j = 1, \ldots, \lceil |\mathcal{C}|/m \rceil$. Comparing documents with similar relevance scores yields higher information gain, as their relative order is more uncertain. We report random partitioning strategies in ablation experiments.

**Score refinement (Line 5).** Each group $\mathcal{B}_j$ is passed to the reranker $g$, and the resulting rankings update TrueSkill parameters. Each outcome is interpreted as a multiplayer game among documents in $\mathcal{B}_j$, where each document is assigned a rank based on its position in the reranked list. We apply the TrueSkill update using listwise ranking as input, using a uniform monotonic ranking score (e.g., rank-based weights) to drive the update. Documents ranked higher than expected receive an increased $\mu_i$, while those ranked lower experience a decreased $\mu_i$. The magnitude reflects how surprising the outcome is relative to prior beliefs. This iterative process gradually sharpens the model's belief in document relevance.

**Stopping criteria (Line 6).** Reranking continues until one or more of the following conditions are met: (i) the number of uncertain documents $|\mathcal{C}|$ falls below the threshold $\tau$, where $\tau$ is the threshold at which the number of $|\mathcal{C}|$ satisfies the stopping criterion, (ii) the top-$k$ set remains unchanged for a

fixed number of iterations, or (iii) the number of loops exceed the maximum computation budget. We use condition (i) by default and (iii) for budget-aware variants, with results on (ii) presented for ablation.

**Final ranking.** Documents are ranked by their $\mu_i$ values to produce the output. By selectively reranking uncertain candidates and refining their relevance estimates through iterative feedback, AcuRank improves accuracy while reducing unnecessary computation. This uncertainty-aware approach allocates effort where most needed, focusing on ambiguous cases and avoiding redundant updates to confidently ranked documents. As a result, AcuRank provides both robust and scalable solutions for efficient listwise reranking under limited computation budgets.

## 4 Experimental setup

### 4.1 Evaluation protocol

**Datasets.** We evaluate reranking performance on two widely used retrieval benchmarks. For TREC Deep Learning (TREC-DL) [16], we use six standard tracks: DL19 (43), DL20 (54), DL21 (53), DL22 (76), DL23 (82), and DL-Hard (50) [40]. For BEIR [17], following Sun et al. [11], we select eight representative datasets: TREC-COVID (50), NFCorpus (323), Signal-1M (97), News (57), Robust04 (249), Touché (49), DBPedia (400), and SciFact (300). The numbers in parentheses indicate the number of queries in each dataset. These datasets span a wide range of domains, including web search, scientific literature, news articles, and argumentative retrieval.

**Evaluation metrics.** We measure ranking accuracy using Normalized Discounted Cumulative Gain at rank 10 (NDCG@10) [41], which reflects the quality of top-ranked documents and is the standard evaluation metric for both TREC-DL and BEIR. To assess efficiency, we count the number of reranker calls per query, which serves as a practical proxy for end-to-end latency in real-world deployments. While our evaluation is based on reranker call counts rather than actual latency, AcuRank supports parallel execution across calls of disjoint candidate groups, enabling faster computation compared to sequential reranking steps or improved ranking accuracy without increasing latency.[4]

**Retrievers and rerankers.** We use BM25 with top-100 candidates as the default first-stage retriever, and evaluate three additional settings: BM25 top-1000, to assess scalability with larger candidate pools, along with SPLADE++ED [42] top-100 and Contriever [43] top-100 (using `facebook/contriever-msmarco` with mean pooling), to evaluate performance with different first-stage signals. Since neural retrievers such as SPLADE and Contriever produce scores on a different scale from BM25, we normalize the top-100 first-stage scores to have a mean of 10 and a standard deviation of 1 per query before applying AcuRank. We perform retrieval from the pre-built Pyserini index [44]. For reranking, we use RankZephyr [12] as the default model, and additionally evaluate RankGPT (`gpt-4.1-mini`) to assess generalization across model families. Both rerankers operate over candidate lists of size $m = 20$, formatted as prompts containing the query and document list.

### 4.2 Baselines

We compare AcuRank against the following reranking baselines with fixed computation per query. All experiments were conducted on a single NVIDIA A6000 GPU (48GB VRAM).

**Sliding windows (SW-$x$):** A standard listwise reranking method that processes the candidate list in overlapping windows of size $m$, with each pass starting from the bottom of the list and progressively refining the ordering toward the top. Since SW-$x$ uses a fixed number of reranker calls per query regardless of ranking difficulty, we evaluate multiple pass configurations (SW-1, SW-2, SW-3) to match the computational cost of our method. We use the implementation from the RankLLM codebase,[5] using the same prompt format, hyperparameters, and input size limits.

**TourRank-$x$:** A listwise reranking method that runs $x$ independent multi-stage tournaments over retrieved candidates [14].[6] At each stage, documents are partitioned into groups based on initial

---

[4]In principle, it is also possible to rerank overlapping groups in parallel and aggregate their outputs, which may improve robustness to noisy reranker predictions. We leave this as a promising direction for future work.

[5]https://github.com/castorini/rank_llm

[6]https://github.com/chenyiqun/TourRank

order, then randomly shuffled. At each stage, an LLM selects the most relevant documents from each group, with survivors receiving stage-specific scores. The process repeats for multiple stages (e.g., $100 \rightarrow 50 \rightarrow 20 \rightarrow 10 \rightarrow 5 \rightarrow 2$), and summing scores across $x$ tournaments yields the final ranking. TourRank-1 uses a single tournament, while TourRank-5 employs five tournaments for lower variance at higher computational cost. For a controlled comparison, we preserve the original algorithm but reuse the exact prompt configuration employed for sliding windows and AcuRank.

**TrueSkill-Static (TS-$c$):** A TrueSkill-based baseline that selects candidates solely based on their estimated relevance scores $\mu_i$, without modeling uncertainty. At each stage $j$, the top-$(m \cdot c_j)$ documents are selected and reranked, where $c = [c_1, \ldots, c_s]$ defines the reranker call allocation per stage. This baseline allows us to isolate the effect of uncertainty modeling in our full method.

### 4.3 AcuRank configurations

Building on the general framework described in Section 3.3, we specify the experimental settings used as our default configuration, unless otherwise noted in the ablation studies (Table 4). The hyperparameters were selected based on empirical evaluation on a subset of TREC-DL19 and DL20, then applied consistently across all other datasets to ensure a fair comparison. We also provide an extended robustness analysis on how computation scales with the candidate-set size $n$ and on ranking stability under document addition; see Supplementary Section F.

**Initialization:** We initialize TrueSkill scores based on first-stage retrieval scores. Specifically, we set the mean $\mu_i$ to the raw retrieval score (e.g., BM25 or SPLADE), and the standard deviation to $\sigma_i = \mu_i/3$. This allows the model to prioritize high-confidence documents from the beginning. Analysis on different variance initialization and justification of the design choice are discussed in Supplementary Section B and Table 5.

**Uncertainty threshold:** We select documents whose rank probability $s_i = \mathbb{P}(x_i > t(k))$ falls within the range $(\epsilon, 1 - \epsilon)$. We use $\epsilon = 0.01$ and $k = 10$ in our default configuration, unless noted otherwise in variant-specific settings.

**Partitioning strategy:** When the number of uncertain documents exceeds the reranker capacity $m = 20$, we divide them into equally sized groups using sequential partitioning. Otherwise, we rerank using a single batch. For ablation, we also evaluate a random grouping variant.

**Stopping criterion:** We terminate reranking when the number of uncertain documents falls below $\tau = 10$, or the reranker call budget is exhausted. Alternative stopping criteria, such as top-$k$ stability, are only used in ablation studies.

**Adaptive variants:** To evaluate the effectiveness of our method under different computational scenarios, we introduce variants of AcuRank that reflect distinct trade-offs between efficiency and accuracy. **AcuRank-$x$** is a strict-budget variant that terminates reranking after a predefined number of reranker calls $x$, enabling fair comparison with baselines under matched computational budgets. **AcuRank-{H, HH}** are high-precision variants designed for settings where ranking quality is prioritized over efficiency. AcuRank-H adopts a tighter tolerance parameter $\epsilon = 0.0001$, and AcuRank-HH additionally applies a stricter stopping criterion ($\tau = 5$), enabling more thorough refinement of uncertain candidates. Extended analyses on different choices of uncertainty thresholds are discussed in Supplementary Section D.3 and Table 13.

## 5 Results and analysis

### 5.1 Effectiveness and efficiency vs. fixed baselines

Table 1 reports NDCG@10 scores and reranker call counts across TREC-DL and BEIR datasets. Under comparable reranker budgets, AcuRank consistently outperforms fixed-computation baselines in terms of overall ranking accuracy. For example, AcuRank-9 matches the average call count of SW-1 (8.8) but achieves slightly higher average NDCG@10, showing that uncertainty-aware allocation can yield better performance even with limited computation.

As more computation becomes available, AcuRank continues to outperform. AcuRank uses 19.7 calls on average, between the budgets of SW-2 (17.6) and SW-3 (26.4), yet exceeds both in accuracy. While the gains of Sliding Windows diminish with additional calls, AcuRank shows a notably larger

Table 1: Results in NDCG@10 on TREC-DL and BEIR using top-100 (top) and top-1000 (bottom) documents on BM25 as first-stage retrieval with RankZephyr reranker. The top row in each block corresponds to the initial ranking without reranking, which serves as a lower-bound reference. Note that some queries retrieve fewer than 100 documents with BM25, slightly reducing average # calls.

| Method | TREC-DL | | | | | | BEIR | | | | | | | | | # Calls |
|---|---|---|---|---|---|---|---|---|---|---|---|---|---|---|---|---|
| | DL19 | DL20 | DL21 | DL22 | DL23 | DL-H | COVID | NFC | Signal | News | R04 | Touche | DBP | Scif | Avg. | |
| First-stage retrieval: *BM25 top 100* | Reranker: *RankZephyr-7B* | | | | | | | | | | | | | | | |
| BM25 | 50.6 | 48.0 | 44.6 | 26.9 | 26.2 | 30.4 | 59.5 | 32.2 | 33.0 | 39.5 | 40.7 | 44.2 | 31.8 | 67.9 | 41.1 | 0.0 |
| SW-1 | 74.0 | 70.2 | 69.5 | 51.5 | 44.5 | 38.6 | 84.1 | 36.8 | 32.0 | 52.3 | 54.0 | 32.4 | 44.5 | 75.5 | 54.3 | 8.8 |
| SW-2 | 74.6 | 70.2 | 70.2 | 51.8 | 45.4 | 38.4 | 84.4 | 37.0 | 33.0 | 52.7 | 54.4 | 31.8 | 44.4 | 75.1 | 54.5 | 17.6 |
| SW-3 | 74.4 | 71.1 | 70.6 | 52.1 | 45.4 | 39.5 | 84.4 | 37.2 | 32.0 | 52.1 | 54.4 | 32.0 | 44.5 | 74.9 | 54.6 | 26.4 |
| TourRank-1 | 74.2 | 68.2 | 69.6 | 51.1 | 45.2 | 38.1 | 81.8 | 36.5 | 30.7 | 51.9 | 54.5 | 31.2 | 43.2 | 71.3 | 53.4 | 12.7 |
| TourRank-2 | 74.1 | 71.9 | 70.8 | 52.3 | 47.1 | 40.3 | 82.5 | 36.8 | 32.2 | 51.8 | 56.5 | 33.4 | 44.0 | 74.3 | 54.9 | 25.5 |
| TourRank-5 | 74.9 | 72.0 | 71.8 | 52.6 | 47.9 | 40.0 | 83.3 | 36.8 | 31.0 | 52.0 | 57.2 | 31.9 | 45.0 | 75.3 | 55.1 | 63.7 |
| TourRank-10 | 74.9 | 71.8 | 71.4 | 53.3 | 47.7 | 39.9 | 83.2 | 37.1 | 31.1 | 53.3 | 57.1 | 32.1 | 44.8 | 75.1 | 55.2 | 127.4 |
| AcuRank-9 | 73.3 | 71.4 | 70.1 | 50.1 | 45.3 | 39.5 | 83.2 | 36.8 | 30.8 | 53.0 | 56.0 | 37.1 | 44.7 | 73.3 | 54.6 | 8.8 |
| AcuRank | 74.2 | 71.8 | 70.3 | 52.0 | 47.0 | 39.4 | 85.3 | 37.2 | 31.8 | 53.9 | 56.6 | 36.5 | 46.0 | 75.3 | **55.5** | 19.7 |
| AcuRank-H | 74.6 | 70.8 | 70.5 | 52.2 | 47.3 | 40.4 | 85.8 | 37.4 | 32.1 | 53.7 | 56.8 | 37.5 | 46.0 | 75.4 | 55.7 | 41.7 |
| AcuRank-HH | 74.7 | 71.8 | 70.6 | 51.9 | 47.0 | 40.0 | 86.1 | 37.5 | 31.3 | 54.4 | 57.8 | 36.1 | 46.0 | 75.4 | 55.8 | 57.2 |
| First-stage retrieval: *BM25 top 1000* | Reranker: *RankZephyr-7B* | | | | | | | | | | | | | | | |
| BM25 | 50.6 | 48.0 | 44.6 | 26.9 | 26.2 | 30.4 | 59.5 | 32.2 | 33.0 | 39.5 | 40.7 | 44.2 | 31.8 | 67.9 | 41.1 | 0.0 |
| SW-1 | 75.1 | 78.8 | 71.5 | 57.5 | 49.5 | 40.9 | 80.7 | 38.0 | 28.9 | 51.0 | 48.0 | 30.9 | 48.0 | 76.4 | 56.2 | 94.6 |
| TourRank-1 | 75.4 | 76.6 | 71.7 | 56.6 | 49.8 | 42.1 | 82.7 | 36.6 | 29.9 | 50.9 | 59.9 | 33.0 | 45.5 | 72.8 | 56.0 | 117.1 |
| AcuRank | 76.7 | 75.3 | 73.1 | 59.3 | 53.5 | 41.0 | 85.0 | 37.0 | 30.7 | 56.5 | 63.2 | 36.2 | 48.8 | 76.0 | **58.0** | **68.4** |

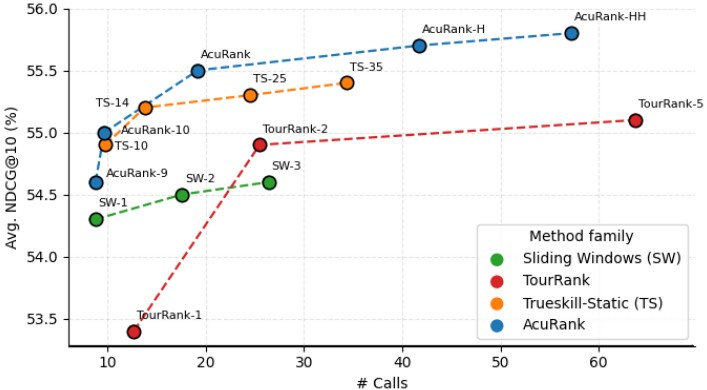

Figure 2: Pareto curves showing the trade-off between accuracy (NDCG@10) and efficiency (# of reranker calls) across reranking methods.

Figure 3: Correlation between WIG-based query difficulty and # reranker calls (Section. 5.3)

improvement from AcuRank-9 to full AcuRank despite a similar increase in cost, highlighting the advantage of adaptively allocating computation based on uncertainty.

Beyond the number of reranker calls, we further evaluate efficiency on: (1) average window size, (2) input length, (3) end-to-end latency, and (4) floating-point operations (FLOPs), as reported in Table 2. Under comparable reranker budgets, AcuRank exhibits consistently higher efficiency across these measures, offering a more faithful assessment than call counts alone. In fact, the number of reranker calls disadvantages AcuRank over baselines. Since AcuRank adaptively partitions uncertain candidates with a maximum window size of $C=20$, this leads to shorter inputs with smaller context numbers in some stages. Results in Table 2 confirm that AcuRank achieves faster runtime and lower input length despite operating under a similar number of calls, with up to 20% reduction in input length and latency on DL19/20, and 15–30% fewer FLOPs on average.

Figure 2 visualizes the accuracy–efficiency trade-off across systems. AcuRank consistently lies along the Pareto frontier, achieving stronger accuracy at a given budget or using fewer calls to reach the same target. In contrast, TourRank attains relatively strong performance only at much higher cost, resulting in a less favorable trade-off.

Table 2: Efficiency analysis of AcuRank vs. Sliding Window baselines, on two different setups, under a similar number of reranking calls. We compared with SW-3 instead of SW-2 for COVID* to match the number of calls, and all efficiency scores are averaged per query. AcuRank demonstrates better efficiency across different measures.

| Metric | SW-2 (left) vs. AcuRank (right) | | | | SW-1 (left) vs. AcuRank-9 (right) | | | |
|---|---|---|---|---|---|---|---|---|
| | DL19 | DL20 | COVID* | News | DL19 | DL20 | COVID | News |
| NDCG@10 ($\uparrow$) | **74.6** / 74.2 | 70.2 / **71.8** | 84.4 / **85.3** | 52.7 / **53.9** | **74.0** / 73.3 | 70.2 / **71.4** | **84.1** / 83.2 | 52.3 / **53.0** |
| # Calls ($\downarrow$) | **18.0** / 18.2 | 18.0 / **16.3** | 27.0 / **21.8** | **18.0** / 18.5 | 9.0 / **8.9** | 9.0 / **8.9** | **9.0** / 9.0 | **9.0** / 9.0 |
| Window size ($\downarrow$) | 20.0 / **16.6** | 20.0 / **16.7** | 20.0 / **16.3** | 20.0 / **16.2** | 20.0 / **18.9** | 20.0 / **18.9** | 20.0 / **18.8** | 20.0 / **18.9** |
| Input length ($\downarrow$) | 2156 / **1831** | 2162 / **1822** | 3991 / **3757** | 3992 / **3989** | 2149 / **2058** | 2152 / **2062** | 3990 / **3865** | 3992 / **3987** |
| Latency (s) ($\downarrow$) | 126 / **106** | 126 / **93** | 213 / **142** | 179 / **150** | 93 / **58** | 92 / **59** | 73 / **67** | 88 / **84** |
| petaFLOPs ($\downarrow$) | 0.61 / **0.52** | 0.62 / **0.46** | 1.8 / **1.3** | 1.2 / **1.2** | 0.30 / **0.29** | 0.31 / **0.29** | 0.59 / **0.57** | 0.59 / 0.59 |

Finally, AcuRank supports flexible deployment under varying computational budgets. Its high-compute variants, AcuRank-H and AcuRank-HH, apply stricter uncertainty thresholds and use more reranker calls, yielding incremental but consistent gains without early saturation. This behavior suggests that AcuRank can function as an anytime prediction method, where users can select a variant according to their available test-time budget and desired accuracy–efficiency trade-off.

## 5.2 Robustness across retrieval scenarios

Table 3 shows that AcuRank maintains strong performance across varying retrieval settings with different first-stage retrievers and reranker models. AcuRank consistently outperforms the fixed-computation baseline SW-1 under comparable reranker call budgets. With SPLADE++ED as the first-stage retriever, AcuRank achieves higher accuracy than SW-3 while using a similar number of calls as SW-1, effectively leveraging strong initial rankings to improve performance without incurring high cost. AcuRank also generalizes well to different reranking models. When using RankGPT (gpt-4.1-mini) instead of RankZephyr, it still achieves higher accuracy than SW-1, suggesting that uncertainty-guided allocation remains effective regardless of the underlying reranker.

## 5.3 Adaptive allocation behavior

We investigate how our uncertainty-guided method adaptively allocates computation across queries and whether this behavior contributes to performance gains. To isolate the effect of adaptive allocation, we compare AcuRank to TrueSkill-Static, a static variant that uses the same probabilistic scoring but selects candidates solely based on estimated relevance $\mu_i$, without considering uncertainty. As shown in Figure 2, TrueSkill-Static outperforms both Sliding Windows and TourRank in terms of accuracy–efficiency trade-off. Nonetheless, AcuRank achieves even higher accuracy with fewer or comparable reranker calls, demonstrating that ranking uncertainty is critical in further improving performance beyond static scheduling.

To further examine how AcuRank adjusts computation at the query level, we analyze the correlation between reranker call counts and query difficulty measured by Weighted Information Gain (WIG) [45], over 612 queries on TREC and BEIR. WIG is computed from BM25 top-100 scores using a $\log(1+s)$ transform with window size $k = 50$, where lower values indicate more ambiguous relevance distributions. Figure 3 shows a significant negative correlation between WIG and the number of reranker calls issued by AcuRank ($\rho = -0.27$, $p < 10^{-8}$), confirming that *more compute is allocated to harder queries*. This adaptive behavior is further detailed in Supplementary Section E, comparing with the Sliding Window variant, where we observe that AcuRank allocates more computation to difficult queries while reducing computation on easier ones. We also report a qualitative example of Bayesian update in the Supplementary Section E.3.

## 5.4 Impact of design choices

Table 4 presents an ablation study evaluating how individual design choices in AcuRank affect its overall effectiveness. Initializing TrueSkill scores from first-stage retrieval improves accuracy over default initialization (55.5 vs. 54.8), though at the cost of more reranker calls. Sequential partitioning outperforms random grouping in both accuracy and efficiency, confirming that comparing documents with similar relevance estimates yields more informative updates. For the stopping criterion, uncertainty-based termination achieves similar accuracy to the top-$k$-unchanged strategy

Table 3: Results in NDCG@10 on BEIR using top-100 documents from (top) SPLADE++ED and (middle) Contriever first-stage retrieval, and RankZephyr reranker and (bottom) BM25 first-stage retrieval and RankGPT (gpt-4.1-mini) reranker. The top row in each block corresponds to the initial ranking without reranking, which serves as a lower-bound reference. Note that some queries retrieve fewer than 100 documents with BM25, slightly reducing average # calls.

| Method | COVID | NFC | Signal | News | R04 | Touche | DBP | Scif | Avg. | # Calls |
|---|---|---|---|---|---|---|---|---|---|---|
| First-stage retrieval: *SPLADE++ED top 100* \| Reranker: *RankZephyr-7B* | | | | | | | | | | |
| SPLADE++ED | 71.1 | 34.5 | 29.6 | 39.4 | 45.8 | 24.4 | 44.1 | 69.9 | 44.8 | 0.0 |
| SW-1 | 85.2 | 37.5 | 29.7 | 50.1 | 62.0 | 28.9 | 49.3 | 75.7 | 52.3 | 9.0 |
| SW-2 | 85.6 | 37.8 | 28.9 | 51.3 | 62.8 | 29.2 | 49.6 | 75.5 | 52.6 | 18.0 |
| SW-3 | 85.2 | 37.7 | 28.8 | 51.3 | 62.9 | 30.1 | 49.7 | 75.4 | 52.6 | 27.0 |
| TourRank-1 | 82.5 | 37.4 | 28.5 | 51.5 | 60.7 | 29.7 | 48.9 | 73.7 | 51.6 | 13.0 |
| TourRank-2 | 84.6 | 37.6 | 29.5 | 52.5 | 61.7 | 28.4 | 50.0 | 74.0 | 52.3 | 26.0 |
| AcuRank | 86.2 | 38.1 | 28.6 | 53.1 | 64.0 | 32.7 | 51.3 | 76.8 | **53.8** | 20.9 |
| First-stage retrieval: *Contriever top 100* \| Reranker: *RankZephyr-7B* | | | | | | | | | | |
| Contriever | 48.0 | 32.0 | 23.8 | 35.3 | 37.7 | 21.7 | 33.0 | 65.1 | 33.1 | 0.0 |
| SW-1 | 70.8 | 38.3 | 26.5 | 49.9 | 54.9 | 30.4 | 46.1 | 75.4 | 45.3 | 9.0 |
| SW-2 | 72.3 | 38.7 | 26.8 | 50.4 | 55.0 | 31.0 | 46.8 | 75.3 | 45.8 | 18.0 |
| SW-3 | 72.6 | 38.6 | 26.6 | 50.5 | 55.0 | 31.3 | 46.9 | 75.4 | 45.9 | 27.0 |
| TourRank-1 | 67.5 | 36.8 | 25.8 | 49.3 | 54.6 | 28.2 | 45.5 | 72.2 | 44.0 | 13.0 |
| TourRank-5 | 70.9 | 38.1 | 26.4 | 52.2 | 57.6 | 30.1 | 47.0 | 74.4 | 46.0 | 65.0 |
| AcuRank | 74.4 | 39.1 | 27.1 | 53.0 | 57.2 | 33.1 | 47.9 | 75.2 | **47.4** | 20.1 |
| First-stage retrieval: *BM25 top-100* \| Reranker: *RankGPT (gpt-4.1-mini)* | | | | | | | | | | |
| BM25 | 59.5 | 32.2 | 33.0 | 39.5 | 40.7 | 44.2 | 31.8 | 67.9 | 43.6 | 0.0 |
| SW-1 | 84.5 | 37.9 | 33.3 | 49.5 | 61.5 | 35.7 | 46.3 | 78.7 | 53.4 | 8.8 |
| AcuRank | 86.4 | 38.3 | 34.8 | 52.1 | 63.0 | 31.9 | 46.2 | 77.2 | **53.7** | 20.8 |

Table 4: Ablation study on **AcuRank**'s design choices with average NDCG@10 on TREC, BEIR, and all datasets. **Init** ($\checkmark$ / $\times$): Initialize TrueSkill scores using first-stage scores vs. default values. **Partitioning**: Method for grouping uncertain documents (sequential order vs. random). **Stopping Criterion**: Terminate iterations when uncertain documents fall below threshold ($|\mathcal{C}| < 10$) or when top-k rankings stabilize across iterations. "—" indicates the default configuration is used. See Appendix for detailed results on individual datasets.

| Init | Partitioning | Stopping Criterion | TREC | BEIR | All | # Calls |
|---|---|---|---|---|---|---|
| $\checkmark$ | - | - | **59.1** | **52.8** | **55.5** | 19.7 |
| $\times$ | - | - | 59.0 | 51.7 | 54.8 | **13.4** |
| $\checkmark$ | Random | - | 58.8 | 52.7 | 55.3 | 22.6 |
| $\checkmark$ | - | Top-k stability | 58.8 | 52.4 | 55.2 | 22.7 |

while reducing the number of reranker calls (19.7 vs. 22.7). Taken together, these results show that each component of our method meaningfully contributes to the method's effectiveness and that the default configuration offers a strong balance between accuracy and computational efficiency.

# 6 Conclusion

We introduce AcuRank, a novel listwise reranking method that performs adaptive computation guided by uncertainty-aware relevance modeling. Our approach maintains probabilistic relevance estimates using TrueSkill and selectively allocates reranking effort to documents with high ranking uncertainty. By focusing computation on ambiguous candidates, AcuRank consistently outperforms fixed-computation baselines across TREC-DL and BEIR, achieving better accuracy-efficiency trade-offs. Beyond a single algorithm, AcuRank serves as a flexible framework for uncertainty-aware adaptive computation in listwise reranking, providing a principled foundation based on probabilistic modeling. This formulation enables fine-grained control over the reranking process and opens opportunities for future research in uncertainty-guided retrieval.

## Acknowledgement

Gyuwan Kim thanks Prof. Tao Yang and Prof. Xifeng Yan at UC Santa Barbara for their valuable feedback and insightful suggestions throughout the course of this work. Soyoung Yoon thanks Jinwoo Kim and the anonymous reviewers of our NeurIPS 2025 submission for giving constructive feedbacks to improve the quality of our paper.

This work was partly supported by Institute of Information & communications Technology Planning & Evaluation (IITP) grant funded by the Korea government(MSIT) [NO.RS-2021-II211343, Artificial Intelligence Graduate School Program (Seoul National University)], and the National Research Foundation of Korea(NRF) grant funded by the Korea government(MSIT) (No. RS-2024-00414981), and the MSIT(Ministry of Science and ICT), Korea, under the ITRC(Information Technology Research Center) support program(IITP-2025-2020-0-01789) supervised by the IITP(Institute for Information & Communications Technology Planning & Evaluation).

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

# Supplementary Materials for AcuRank: Uncertainty-Aware Adaptive Computation for Listwise reranking

## A Overview

This supplementary material provides additional details and extended results for our AcuRank framework. Specifically:

- **Section B** describes the mathematical foundations of our Bayesian modeling approach, including the TrueSkill update equations and our efficient approximation of cumulative rank distributions. We also report the computational overhead of TrueSkill and additional experimental justification of the variance initialization.

- **Section C** specifies our experimental setup, including hardware specifications, software configurations, datasets, licenses, and modifications to baseline implementations.

- **Section D** presents comprehensive experimental results, including dataset-wise performance metrics omitted from the main paper due to space constraints. This includes extended results for all baselines (Tables 1-3), full numbers for the TrueSkill-Static variants shown in Figure 2, additional experiments with **RankVicuna-7B** and **Llama-3.3-70B-Instruct** demonstrating the generalizability of our approach, and extended analysis on AcuRank's sensitivity to uncertainty thresholds.

- **Section E** provides a deeper analysis of AcuRank's adaptive behavior, including query-level compute allocation patterns and a breakdown of performance on easy versus hard queries. Also, we provide a qualitative example of the Bayesian update of AcuRank.

- **Section F** discusses the AcuRank's robustness among scalability and ranking stability over varying $n$ and document addition.

- **Section G** discusses current limitations and potential future directions.

- **Section H** lists the exact prompt templates used with RankZephyr, RankVicuna, and Llama-3.3-70B-Instruct for listwise reranking.

## B Supplement on Bayesian modeling and score updates in TrueSkill

### B.1 TrueSkill as a *Bayesian* model

TrueSkill [15] models the latent relevance (originally, "skill") of each document $x_i$ as a Gaussian random variable, $x_i \sim \mathcal{N}(\mu_i, \sigma_i^2)$, starting with a prior and updating it iteratively via Bayesian inference. Each listwise reranker output, denoted $g(\mathcal{D}'; \mathcal{M})$, is treated as noisy evidence about pairwise preferences among a subset $\mathcal{D}'$ of documents, where $\mathcal{M}$ is the reranking model. Using belief propagation on a factor graph, TrueSkill analytically updates the posterior distribution to

$$p(x_i \mid g(\mathcal{D}'; \mathcal{M})) = \mathcal{N}(\mu_i', (\sigma_i')^2).$$

We denote the current update round by $t$. At $t = 0$, $(\mu_i^{(0)}, \sigma_i^{(0)})$ represents the prior initialized from first-stage retrieval scores. After observing the reranking output, the posterior is updated to $(\mu_i^{(t+1)}, \sigma_i^{(t+1)})$.

### B.2 Approximating the cumulative rank distribution

We approximate the probability that document $D_i$ ranks within the top $r$ positions as

$$P(r_i \leq r) \approx P(x_i > t(r)), \text{where} \sum_j P(x_j > t(r)) = r.$$

The left–hand side is the probability that at most $r - 1$ other documents outrank $D_i$. The right-hand side approximates this count by evaluating whether $x_i$ exceeds a threshold $t(r)$ calibrated such that the expected number of documents exceeding this threshold equals $r$. This approximation becomes accurate under two assumptions: (1) the latent scores $x_i$ follow independent Gaussian distributions with similar variances, and (2) the number of candidates $n$ is sufficiently large, so that

Table 5: **Ablations on initializing variance** (BM25 top-100, RankZephyr) for AcuRank. $\mu/3$ corresponds to the final configuration used in AcuRank.

| $\sigma$ | DL19 | DL20 | DL21 | DL22 | DL23 | DL-H | COV. | NFC | Sig. | News | R04 | Tou. | DBP | Scif | Avg. | # Calls |
|---|---|---|---|---|---|---|---|---|---|---|---|---|---|---|---|---|
| 8 (fixed) | 74.1 | 72.6 | 70.5 | 51.3 | 46.9 | 39.5 | 85.5 | 36.9 | 31.1 | 52.5 | 56.2 | 33.3 | 45.1 | 76.4 | 55.1 | 13.1 |
| $\mu/2$ | 74.1 | 71.3 | 70.7 | 51.8 | 46.2 | 39.4 | 85.2 | 37.4 | 32.0 | 52.6 | 56.3 | 37.3 | 46.0 | 75.7 | 55.4 | 16.4 |
| $\mu/3$(ours) | 74.2 | 71.8 | 70.3 | 52.0 | 47.0 | 39.4 | 85.3 | 37.2 | 31.8 | 53.9 | 56.6 | 36.5 | 46.0 | 75.3 | **55.5** | **19.2** |

the expected number of exceedances concentrates near its mean by the law of large numbers. This provides an *efficient closed-form* alternative to the expensive $O(n^2)$ dynamic programming approach for computing the full ranking distribution over permutations. Instead of enumerating all possible rankings, we approximate $P(r_i \leq r)$ using a single Gaussian tail probability, $1 - \Phi(\cdot)$.

### B.3 TrueSkill score update equations

The reranking over each batch $\mathcal{B}$ is treated as a *multi-player game*. The observed ranking outcome is given by $g(\mathcal{B}; \mathcal{M})$. For every adjacent document pair in this order, a win–loss factor is defined to model their pairwise preference. Applying message passing on the factor graph yields the following canonical update equations for each document $D_i$:

$$\mu_i^{(t+1)} = \mu_i^{(t)} + \frac{\sigma_i^2}{c}\lambda, \quad (\sigma_i^{(t+1)})^2 = \sigma_i^2 \left(1 - \frac{\sigma_i^2}{c}\upsilon\right),$$

where $c = \sqrt{\sum_{D_j \in \mathcal{B}}(\sigma_j^2 + \beta^2)}$. The values $\lambda$ and $\upsilon$ are closed-form functions derived from the win probability between adjacent items (see Herbrich et al. [15] for a full derivation). Intuitively, when a document ranks higher than expected, its mean $\mu_i$ increases. Unexpected outcomes also reduce $\sigma_i$, reflecting greater confidence in the document's estimated relevance.

**Side note on implementation.** We employ the Python `trueskill` library.[7] The relevance of each candidate passage is represented as a `trueskill.Rating` class. After the listwise reranker outputs an ordering, we update all ratings by calling `trueskill.rate` function with the argument `rating_groups= list of ratings`, and `ranks=[0,1,...,n]`, where $n$ is the number of passages. The function returns a list of posterior ratings containing $(\mu, \sigma)$, which serve as the priors for the next iteration.[8]

### B.4 Proportional uncertainty initialization based on retrieval scores

We initialize each document's relevance score as a Gaussian distribution parameterized by its first-stage retrieval score. Specifically, for each document $D_i$ with retrieval score $s_i$, we set the mean and standard deviation as $\mu_i = s_i$ and $\sigma_i = \mu_i/3$. This initialization preserves the *relative scale* of the retrieval scores while explicitly modeling their uncertainty. A higher retrieval score indicates greater expected relevance ($\mu_i \uparrow$), but it may also be more prone to overestimation. To reflect this, we assign a proportionally larger prior uncertainty ($\sigma_i \uparrow$) to such documents. The $1/3$ ratio follows the standard $\mu{:}\sigma{=}3{:}1$ default used in TrueSkill, and yields a conservative *displayed rating* $R_i = \mu_i - 3\sigma_i = 0$. This provides a non-negative lower bound (semantically natural for relevance), while leaving substantial headroom for upward revision as reranking evidence arrives. Statistically, $[\mu_i - \mu_i,\ \mu_i + \mu_i] = [0, 2\mu_i]$ corresponds to $[\mu_i \pm 3\sigma_i]$ since $3\sigma_i = \mu_i$, enclosing $\approx 99.7\%$ of Gaussian mass; thus the prior is cautious yet flexible, centered on the retrieval signal but amenable to change. Using a larger multiple (e.g., $\sigma_i = \mu_i/2$) makes the prior overly diffuse and weakens the retrieval signal, whereas a smaller multiple would risk overconfidence and slower correction; the three-sigma choice strikes a practical middle ground that is widely adopted in TrueSkill deployments.

**Experimental justification.** To further support this design, we conduct an ablation study comparing three initialization variants: (1) a fixed uncertainty ($\sigma{=}8$), (2) a more conservative variant ($\sigma_i = \mu_i/2$), and (3) our default setting ($\sigma_i = \mu_i/3$). As shown in Table 5, the default configuration performs best overall, while all variants preserve the same qualitative trends. This indicates that AcuRank is robust to the specific choice of initialization and supports the effectiveness of the three-sigma prior.

---

[7] https://github.com/sublee/trueskill
[8] https://trueskill.org/#trueskill.TrueSkill.rate

Table 6: Comparing **per-query** end-to-end latency with the TrueSkill computational overhead, on four selected benchmarks.

| | DL19 (AcuRank-9) | DL19 (AcuRank) | DL20 (AcuRank-9) | DL20 (AcuRank) | COVID (AcuRank-9) | COVID (AcuRank) | News (AcuRank-9) | News (AcuRank) |
|---|---|---|---|---|---|---|---|---|
| Total (s) | 58.28 | 105.98 | 59.28 | 93.30 | 67.32 | 142.08 | 83.65 | 149.58 |
| TrueSkill (s) | 0.06 | 0.11 | 0.06 | 0.10 | 0.06 | 0.13 | 0.06 | 0.11 |
| TrueSkill ratio (%) | 0.10% | 0.10% | 0.10% | 0.11% | 0.09% | 0.09% | 0.07% | 0.08% |

### B.5 Score refinement (Algorithm 1, Line 5)

After each reranker call, we perform the following update procedure:

1. Convert the returned ranking over the current batch $B$ into a set of pairwise win–loss observations.

2. Run one round of TrueSkill message passing to update each document's parameters $(\mu_i, \sigma_i)$ in $B$ based on these observations.

3. Recompute the ranking-uncertainty scores $s_i = P(x_i > t(k))$ to guide selection in the next iteration.

This iterative posterior refinement progressively sharpens the relevance estimates. As a result, the uncertainty values $\sigma_i$ naturally decrease over time. Documents whose ranks stabilize early tend to exit the *uncertain* set, allowing us to avoid unnecessary reranker calls and reduce overall computation.

### B.6 Computational overhead of TrueSkill

We additionally measured and reported the computational overhead of TrueSkill at Table 6, which was **negligible** compared to the cost of LLM reranker calls. On the DL19 benchmark, the total runtime of AcuRank was approximately 58.28 seconds, among which the cumulative time spent on TrueSkill updates was only 0.06 seconds (0.1%). The portion of running Trueskill versus the total process was consistent (0.1%) across subsets of data (DL19, DL20, TREC-COVID, News) and for AcuRank-9 and AcuRank. This was measured using `time.time()` around the call to `trueskill.rate()`, which can be further observed at the official codebase. The `trueskill.rate()` function performs lightweight updates to the mean and variance of document scores based on closed-form Gaussian updates, which are efficiently computed on CPU. This confirms that nearly all computation time in AcuRank is spent on LLM calls, with minimal overhead from the uncertainty modeling.

## C Implementation Details

### C.1 Hardware specifications and software configurations

**Compute nodes:** Experiments were conducted on either (i) an ASUS ESC8000-E11 server equipped with dual 4[th]-Gen Intel Xeon processors, 64 CPU threads, 1.1 TB RAM, and eight NVIDIA A6000 GPUs (48 GB each), or (ii) a workstation with eight RTX 3090 GPUs (24 GB each).

**Inference stack:** All LLM inference was performed using the `transformers` library, without acceleration backends such as vLLM or FastChat. Greedy decoding was used throughout, with a fixed random seed to ensure reproducibility.

**Reproducibility:** We release a complete end-to-end reproduction script, packaged as a `.zip` archive, which replicates reported experiments.

**Token limits:** Following the RankLLM codebase[9], all inputs were truncated to a maximum of 4,096 tokens.

**Default hyperparameters:** Unless otherwise specified, we adopt the default hyperparameter settings provided by the RankLLM implementation.

---

[9]`https://github.com/castorini/rank_llm`

### C.2 Datasets and licenses

**Abbreviations.** For brevity, we use the following shorthands throughout our tables: DL19–DL23 (TREC-DL 2019–2023), DL-H (DL-Hard), COVID (TREC-COVID), NFC (NFCorpus), Signal (Signal-1M), News (TREC-News), R04 (Robust04), Touche (Webis-Touche2020), DBP (DBPedia), and Sci (SciFact).

**TREC-DL (2019–2023 tracks).** The Topics and Qrels are in the public domain as U.S. Government work.[10] The underlying corpus (MS MARCO + ORCAS) is released by Microsoft for **non-commercial research purposes only**, with no IP rights to the documents and "as-is" usage at the user's own risk.[11]

**DL-HARD [40].** We use the annotation files and BM25 retrieval result files hosted on GitHub (`https://github.com/grill-lab/DL-Hard`). The repository does not specify a formal license. Following author guidance and consistent with MS MARCO terms (see footnote 11), we use the data strictly for non-commercial academic research.

**BEIR [17].** The framework code for BEIR is under the Apache 2.0 license.[12] Each constituent dataset retains the license of its original source (e.g., public-domain for TREC-COVID). For our experiments, we use pre-computed BM25 and SPLADE top-1000 retrieval runs (included in our anonymized supplementary code), which are released under the BigScience OpenRAIL-M license.

**TrueSkill rating algorithm.** We rely on the open-source Python package `trueskill` (v0.4.5)[13], distributed under the permissive **BSD 3-Clause License**. The *TrueSkill*[TM] trademark and the original Bayesian rating system remain the property of Microsoft, who permits the name and algorithm to be used only in Xbox Live titles or *non-commercial* projects.[14] Our usage complies with these terms, as it is strictly limited to non-profit academic research.

### C.3 Evaluation protocol

All reported metric values are **rounded to one decimal place** for consistency. For each method, we first compute the mean number of reranker calls *within each dataset*. Specifically, we calculate the average across all queries belonging to a given dataset, resulting in one mean value per dataset. We then compute the overall value reported in our tables by taking a simple unweighted average of these 14 per-dataset means. This ensures that each dataset contributes equally, regardless of the number of queries it contains. Note that this is a dataset-level **macro average**, not a micro average computed over all individual queries pooled together.

### C.4 Sliding windows baseline

The sliding windows baselines rerank the top-100 candidates using fixed-size windows of size 20 and a stride of 10. This configuration results in *nine* overlapping windows per query. When the list length is not divisible by 20, we form a final, smaller window containing the remaining documents. This ensures full coverage of all retrieved candidates.

### C.5 TourRank baseline

We adapted the public TourRank implementation[15] to handle candidate sets with fewer than 100 passages. The original code assumes $|\mathcal{D}| = 100$ and executes five tournament stages. However, for some queries in certain datasets (e.g., NFCorpus), the query text is extremely short, so BM25 retrieves fewer than 100 candidate documents ($|\mathcal{D}| < 100$). To handle such cases, we added the simple stage-skip rule below and a one-line boundary check to the original grouping function, in order to avoid redundant inference and out-of-range errors. Our modifications are twofold:

---

[10]`https://trec.nist.gov`
[11]`https://microsoft.github.io/msmarco/`
[12]`https://github.com/beir-cellar/beir`
[13]`https://github.com/sublee/trueskill`
[14]`https://trueskill.org/`
[15]`https://github.com/chenyiqun/TourRank`

Table 7: Full results corresponding to Table 1 of the main paper, with annotated `avg_calls` per dataset. Only SW-**1** and TourRank-**1** are shown because these variants used fixed computation per query; the counts for SW-**N** or TourRank-**N** for $N > 1$ can be obtained by multiplying the reported numbers by $N$. Slightly lower values on NFCorpus, DBPedia, and SciFact reflect queries that retrieve fewer than 100 documents with BM25, which proportionally reduces the average number of calls.

| Method | TREC-DL | | | | | | BEIR | | | | | | | | Avg. | # Calls |
|---|---|---|---|---|---|---|---|---|---|---|---|---|---|---|---|---|
| | DL19 | DL20 | DL21 | DL22 | DL23 | DL-H | COVID | NFC | Signal | News | R04 | Touche | DBP | Scif | | |
| First-stage retrieval: *BM25 top 100* | Reranker: *RankZephyr-7B* | | | | | | | | | | | | | | | |
| BM25 (no rerank) | 50.6 | 48.0 | 44.6 | 26.9 | 26.2 | 30.4 | 59.5 | 32.2 | 33.0 | 39.5 | 40.7 | 44.2 | 31.8 | 67.9 | 41.1 | 0.0 |
| ↪ avg_calls | 0 | 0 | 0 | 0 | 0 | 0 | 0 | 0 | 0 | 0 | 0 | 0 | 0 | 0 | 0.0 | 0.0 |
| SW-1 | 74.0 | 70.2 | 69.5 | 51.5 | 44.5 | 38.6 | 84.1 | 36.8 | 32.0 | 52.3 | 54.0 | 32.4 | 44.5 | 75.5 | 54.3 | 8.8 |
| ↪ avg_calls | 9.0 | 9.0 | 9.0 | 9.0 | 9.0 | 9.0 | 9.0 | 6.2 | 9.0 | 9.0 | 9.0 | 9.0 | 9.0 | 9.0 | 8.8 | 8.8 |
| TourRank-1 | 74.2 | 68.2 | 69.6 | 51.1 | 45.2 | 38.1 | 81.8 | 36.5 | 30.7 | 51.9 | 54.5 | 31.2 | 43.2 | 71.3 | 53.4 | 12.7 |
| ↪ avg_calls | 13.0 | 13.0 | 13.0 | 13.0 | 13.0 | 13.0 | 13.0 | 9.3 | 13.0 | 13.0 | 13.0 | 13.0 | 13.0 | 13.0 | 12.7 | 12.7 |
| AcuRank-9 | 73.3 | 71.4 | 70.1 | 50.1 | 45.3 | 39.5 | 83.2 | 36.8 | 30.8 | 53.0 | 56.0 | 37.1 | 44.7 | 73.3 | 54.6 | 8.8 |
| ↪ avg_calls | 9.0 | 8.9 | 9.0 | 9.0 | 9.0 | 9.0 | 9.0 | 6.7 | 9.0 | 9.0 | 9.0 | 8.9 | 8.9 | 8.9 | 8.8 | 8.8 |
| AcuRank | 74.2 | 71.8 | 70.3 | 52.0 | 47.0 | 39.4 | 85.3 | 37.2 | 31.8 | 53.9 | 56.6 | 36.5 | 46.0 | 75.3 | 55.5 | 19.7 |
| ↪ avg_calls | 18.2 | 16.3 | 15.5 | 18.7 | 17.4 | 16.3 | 21.8 | 23.3 | 30.4 | 18.5 | 17.8 | 19.4 | 21.5 | 20.1 | 19.7 | 19.7 |
| AcuRank-H | 74.6 | 70.8 | 70.5 | 52.2 | 47.3 | 40.4 | 85.8 | 37.4 | 32.1 | 53.7 | 56.8 | 37.5 | 46.0 | 75.4 | 55.7 | 41.7 |
| ↪ avg_calls | 36.9 | 35.3 | 29.6 | 40.0 | 36.6 | 32.9 | 43.9 | 51.5 | 58.8 | 38.9 | 37.6 | 44.2 | 48.4 | 49.0 | 41.7 | 41.7 |
| AcuRank-HH | 74.7 | 71.8 | 70.6 | 51.9 | 47.0 | 40.0 | 86.1 | 37.5 | 31.3 | 54.4 | 57.8 | 36.1 | 46.0 | 75.4 | 55.8 | 57.2 |
| ↪ avg_calls | 53.3 | 46.6 | 43.4 | 60.6 | 60.2 | 48.9 | 60.2 | 59.8 | 75.2 | 52.2 | 50.6 | 57.0 | 68.6 | 63.9 | 57.2 | 57.2 |
| First-stage retrieval: *BM25 top 1000* | Reranker: *RankZephyr-7B* | | | | | | | | | | | | | | | |
| BM25 | 50.6 | 48.0 | 44.6 | 26.9 | 26.2 | 30.4 | 59.5 | 32.2 | 33.0 | 39.5 | 40.7 | 44.2 | 31.8 | 67.9 | 41.1 | 0.0 |
| ↪ avg_calls | 0 | 0 | 0 | 0 | 0 | 0 | 0 | 0 | 0 | 0 | 0 | 0 | 0 | 0 | 0.0 | 0.0 |
| SW-1 | 75.1 | 78.8 | 71.5 | 57.5 | 49.5 | 40.9 | 80.7 | 38.0 | 28.9 | 51.0 | 48.0 | 30.9 | 48.0 | 76.4 | 56.2 | 94.6 |
| ↪ avg_calls | 99.0 | 99.0 | 99.0 | 99.0 | 99.0 | 99.0 | 99.0 | 40.1 | 99.0 | 99.0 | 99.0 | 99.0 | 98.4 | 97.0 | 94.6 | 94.6 |
| AcuRank | 76.7 | 75.3 | 73.1 | 59.3 | 53.5 | 41.0 | 85.0 | 37.0 | 30.7 | 56.5 | 63.2 | 36.2 | 48.8 | 76.0 | **58.0** | **68.4** |
| ↪ avg_calls | 67.8 | 65.6 | 67.6 | 74.4 | 72.0 | 68.8 | 72.8 | 40.6 | 82.2 | 68.0 | 68.7 | 70.4 | 74.2 | 64.7 | 68.4 | 68.4 |

Table 8: Results in NDCG@10 on BEIR with two alternative first-stage / reranker configurations, corresponding to Table 2 of our main paper. For every method that performs reranking, the dataset-wise average number of passes is given in the immediately following row (prefixed by ↪).

| Method | COVID | NFC | Signal | News | R04 | Touche | DBP | Scif | Avg. | # Calls |
|---|---|---|---|---|---|---|---|---|---|---|
| First-stage retrieval: *SPLADE++ED top-100* | Reranker: *RankZephyr-7B* | | | | | | | | | |
| SPLADE++ED | 71.1 | 34.5 | 29.6 | 39.4 | 45.8 | 24.4 | 44.1 | 69.9 | 44.8 | 0.0 |
| SW-1 | 85.2 | 37.5 | 29.7 | 50.1 | 62.0 | 28.9 | 49.3 | 75.7 | 52.3 | 9.0 |
| ↪ avg_calls | 9.0 | 9.0 | 9.0 | 9.0 | 9.0 | 9.0 | 9.0 | 9.0 | 9.0 | 9.0 |
| SW-2 | 85.6 | 37.8 | 28.9 | 51.3 | 62.8 | 29.2 | 49.6 | 75.5 | 52.6 | 18.0 |
| SW-3 | 85.2 | 37.7 | 28.8 | 51.3 | 62.9 | 30.1 | 49.7 | 75.4 | 52.6 | 27.0 |
| **AcuRank** | 86.2 | 38.1 | 28.6 | 53.1 | 64.0 | 32.7 | 51.3 | 76.8 | **53.8** | 20.9 |
| ↪ avg_calls | 17.7 | 17.4 | 28.4 | 28.4 | 18.5 | 18.1 | 20.4 | 18.2 | 20.9 | 20.9 |
| First-stage retrieval: *BM25 top-100* | Reranker: *RankGPT* (`gpt-4.1-mini`) | | | | | | | | | |
| BM25 | 59.5 | 32.2 | 33.0 | 39.5 | 40.7 | 44.2 | 31.8 | 67.9 | 43.6 | 0.0 |
| SW-1 | 84.5 | 37.9 | 33.3 | 49.5 | 61.5 | 35.7 | 46.3 | 78.7 | 53.4 | 8.6 |
| ↪ avg_calls | 9.0 | 6.2 | 9.0 | 9.0 | 9.0 | 9.0 | 9.0 | 9.0 | 8.6 | 8.6 |
| **AcuRank** | 86.4 | 38.3 | 34.8 | 52.1 | 63.0 | 31.9 | 46.2 | 77.2 | **53.7** | 20.8 |
| ↪ avg_calls | 20.9 | 25.0 | 22.0 | 18.3 | 18.8 | 19.4 | 20.0 | 21.7 | 20.8 | 20.8 |

**Stage skip rule.** We skip Stages 1 through 4 when $|\mathcal{D}|$ is less than or equal to 50, 20, 10, and 5, respectively. When $|\mathcal{D}| \leq 2$, we return the BM25 order directly.

**Boundary check (grouping).** We retain the original index modulo grouping logic from the public implementation and insert a simple boundary check (`if idx < len(D)`) to prevent `IndexError`.

# D Extended experimental results

## D.1 Average number of reranker calls per dataset

We report the number of reranker calls per dataset with NDCG@10 scores, which correspond to the Tables and Figures reported in the main paper. Table 7 reports the NDCG@10 scores for each dataset,

Table 9: Dataset-wise results in NDCG@10 of *TrueSkill-Static* with different configurations $c$ (BM25 top-100 + RankZephyr-7B), corresponding to Figure 2 of the main paper. TS-$c$ is represented as TS-$N$, where $N = \sum_i c_i$, for short. The last two columns are the macro average over 14 datasets and the macro mean number of reranker calls.

| Method | TREC-DL | | | | | | BEIR | | | | | | | | Avg. | # Calls |
|---|---|---|---|---|---|---|---|---|---|---|---|---|---|---|---|---|
| | DL19 | DL20 | DL21 | DL22 | DL23 | DL-H | COVID | NFC | Signal | News | R04 | Touche | DBP | Scif | | |
| TS-10 [5-2-2-1] | 74.4 | 71.2 | 69.9 | 50.7 | 45.7 | 39.8 | 82.6 | 37.0 | 31.9 | 52.5 | 55.8 | 37.2 | 45.4 | 74.6 | 54.9 | 9.8 |
| TS-14 [5-3-3-3] | 75.0 | 71.8 | 70.2 | 51.1 | 45.6 | 40.5 | 83.6 | 37.0 | 31.3 | 53.8 | 56.2 | 36.7 | 45.7 | 74.8 | 55.2 | 13.8 |
| TS-25 [5-5-5-5-5] | 74.8 | 71.8 | 70.3 | 51.3 | 45.7 | 40.4 | 83.5 | 37.1 | 32.0 | 53.1 | 56.7 | 36.8 | 45.6 | 75.1 | 55.3 | 24.5 |
| TS-25 [5-4-4-4-4] | 75.0 | 71.6 | 70.5 | 51.3 | 46.0 | 39.9 | 84.2 | 37.1 | 32.0 | 53.2 | 56.5 | 36.0 | 45.6 | 75.1 | 55.3 | 24.5 |
| TS-35 [5-3-3-3-3-3-3-3-3-3] | 75.5 | 72.0 | 70.7 | 51.5 | 45.8 | 40.1 | 84.4 | 37.0 | 31.1 | 53.9 | 56.1 | 35.9 | 45.3 | 75.8 | 55.4 | 34.4 |

Table 10: Dataset-wise NDCG@10 for the **AcuRank** design-choice ablation, corresponding to Table 3 of our main paper. For every variant the second line (prefixed by $\hookrightarrow$) reports the dataset-specific average number of passes.

| Variant | TREC-DL | | | | | | BEIR | | | | | | | | Avg. | # Calls |
|---|---|---|---|---|---|---|---|---|---|---|---|---|---|---|---|---|
| | DL19 | DL20 | DL21 | DL22 | DL23 | DL-H | COVID | NFC | Signal | News | R04 | Touche | DBP | Scif | | |
| AcuRank (default) | 74.2 | 71.8 | 70.3 | 52.0 | 47.0 | 39.4 | 85.3 | 37.2 | 31.8 | 53.9 | 56.6 | 36.5 | 46.0 | 75.3 | **55.5** | 19.7 |
| $\hookrightarrow$ avg_calls | 18.2 | 16.3 | 15.5 | 18.7 | 17.4 | 16.3 | 21.8 | 23.3 | 30.4 | 18.5 | 17.8 | 19.4 | 21.5 | 20.1 | 19.7 | 19.7 |
| No 1st-stage init | 74.1 | 70.8 | 70.9 | 52.0 | 47.3 | 38.8 | 84.3 | 36.8 | 30.5 | 52.1 | 55.3 | 33.5 | 44.9 | 75.8 | 54.8 | 13.4 |
| $\hookrightarrow$ avg_calls | 12.5 | 12.7 | 12.3 | 12.5 | 13.3 | 12.9 | 12.9 | 9.6 | 21.4 | 13.8 | 13.3 | 13.0 | 14.1 | 13.4 | 13.4 | 13.4 |
| Random chunking | 74.2 | 71.2 | 70.8 | 51.8 | 47.2 | 37.6 | 87.0 | 37.3 | 31.6 | 53.9 | 56.8 | 34.5 | 46.0 | 74.8 | 55.3 | 22.6 |
| $\hookrightarrow$ avg_calls | 19.9 | 17.4 | 17.4 | 19.6 | 20.7 | 18.1 | 25.7 | 30.4 | 34.4 | 19.6 | 19.6 | 23.4 | 23.7 | 26.8 | 22.6 | 22.6 |
| Top-$k$ stability stop | 75.2 | 69.5 | 70.0 | 51.4 | 46.5 | 40.1 | 84.7 | 37.0 | 31.7 | 52.5 | 56.1 | 37.6 | 45.4 | 74.5 | 55.2 | 22.7 |
| $\hookrightarrow$ avg_calls | 22.3 | 20.3 | 18.0 | 22.8 | 20.7 | 20.1 | 23.7 | 21.6 | 29.3 | 22.1 | 22.2 | 24.4 | 24.5 | 25.3 | 22.7 | 22.7 |

along with the average number of reranker calls. These results complement Table 1 in the main paper by providing dataset-level breakdowns. Additionally, Table 9 reports the complete per-dataset results for TrueSkill-Static with different configurations shown in Figure 2. Table 8 summarizes dataset-wise results for the SPLADE++ED first-stage retrieval results and results with RankGPT (`gpt-4.1-mini`) as a reranker, including the average number of passes, corresponding to Table 3 in the main paper. Finally, Table 10 reports the *per-dataset* average number of reranker calls and the resulting NDCG@10 for each design-choice variant, corresponding with Table 4 in our main paper.

A key observation from these extended results is that, unlike static approaches such as sliding windows or TourRank, our method exhibits varying numbers of calls across different datasets. This adaptive behavior reflects our uncertainty-aware framework's ability to allocate computational resources based on the inherent difficulty and characteristics of each dataset.

## D.2 Generalization across rerankers

To evaluate the generalizability of AcuRank across different reranking models and scales, we conduct comprehensive experiments using both RankVicuna-7B [8] and Llama-3.3-70B-Instruct [46] models.

**Evaluation with RankVicuna-7B.** We first test AcuRank using **RankVicuna-7B** (huggingface identifier of `castorini/rank_vicuna_7b_v1`) on BM25 top-100 retrieval results. As shown in Table 11, while RankVicuna achieves lower absolute NDCG@10 scores than RankZephyr, AcuRank maintains its effectiveness pattern. Specifically, AcuRank-9 achieves +0.7 improvement over SW-1 that uses same number of calls (8.8). Meanwhile, AcuRank achieves +1.8 NDCG@10 improvement over SW-1, and +1.2 improvement over SW-3 despite using fewer calls (19.5 vs 26.4), and +1.5 improvement over TourRank-2 using similar number of calls (25.5 vs 26.4). In summary, AcuRank consistently outperforms both sliding windows and TourRank baselines while requiring fewer reranker calls.

**Evaluation with Llama-3.3 70B as a zero-shot reranker.** To test whether AcuRank continues to yield benefits for *larger, purely zero-shot* rerankers, we replaced our 7B parameter RankZephyr

Table 11: Results in NDCG@10 using BM25 top-100 first-stage retrieval with RankVicuna-7B reranker. For every variant the second line (prefixed by ↪) reports the dataset-specific average number of passes. Lower call counts on some datasets (NFCorpus, DBPedia, SciFact) reflect queries retrieving fewer than 100 documents with BM25.

| Method | TREC-DL | | | | | | BEIR | | | | | | | | Avg. | # Calls |
|---|---|---|---|---|---|---|---|---|---|---|---|---|---|---|---|---|
| | DL19 | DL20 | DL21 | DL22 | DL23 | DL-H | COVID | NFC | Signal | News | R04 | Touche | DBP | Scif | | |
| First-stage retrieval: *BM25 top-100*  \|  Reranker: *RankVicuna-7B* | | | | | | | | | | | | | | | | |
| BM25 | 50.6 | 48.0 | 44.6 | 26.9 | 26.2 | 30.4 | 59.5 | 32.2 | 33.0 | 39.5 | 40.7 | 44.2 | 31.8 | 67.9 | 41.1 | 0.0 |
| ↪ avg_calls | 0 | 0 | 0 | 0 | 0 | 0 | 0 | 0 | 0 | 0 | 0 | 0 | 0 | 0 | 0.0 | 0.0 |
| SW-1 | 66.4 | 62.5 | 60.4 | 40.9 | 38.2 | 35.1 | 77.4 | 33.0 | 33.7 | 45.2 | 46.9 | 33.7 | 43.6 | 69.5 | 49.0 | 8.8 |
| ↪ avg_calls | 9.0 | 9.0 | 9.0 | 9.0 | 9.0 | 9.0 | 9.0 | 6.2 | 9.0 | 9.0 | 9.0 | 9.0 | 9.0 | 9.0 | 8.8 | 8.8 |
| SW-2 | 67.4 | 63.2 | 60.5 | 40.8 | 38.1 | 36.2 | 78.6 | 33.3 | 33.5 | 45.4 | 47.1 | 33.7 | 43.5 | 70.6 | 49.4 | 17.6 |
| ↪ avg_calls | 18.0 | 18.0 | 18.0 | 18.0 | 18.0 | 18.0 | 18.0 | 12.4 | 18.0 | 18.0 | 18.0 | 18.0 | 18.0 | 18.0 | 17.6 | 17.6 |
| SW-3 | 67.5 | 63.2 | 60.6 | 41.0 | 38.2 | 35.9 | 78.7 | 33.5 | 33.9 | 45.6 | 47.2 | 34.8 | 43.9 | 69.9 | 49.6 | 26.4 |
| ↪ avg_calls | 27.0 | 27.0 | 27.0 | 27.0 | 27.0 | 27.0 | 27.0 | 18.6 | 27.0 | 27.0 | 27.0 | 27.0 | 26.9 | 27.0 | 26.4 | 26.4 |
| SW-4 | 67.5 | 63.2 | 60.6 | 41.0 | 37.3 | 36.2 | 78.7 | 33.4 | 33.8 | 45.9 | 47.4 | 34.4 | 43.8 | 70.3 | 49.5 | 35.2 |
| ↪ avg_calls | 36.0 | 36.0 | 36.0 | 36.0 | 36.0 | 36.0 | 36.0 | 24.9 | 36.0 | 36.0 | 36.0 | 36.0 | 35.9 | 36.0 | 35.2 | 35.2 |
| TourRank-1 | 65.7 | 63.4 | 61.2 | 42.5 | 35.4 | 35.5 | 76.7 | 31.6 | 31.6 | 43.9 | 46.7 | 29.2 | 39.8 | 62.2 | 47.5 | 12.7 |
| ↪ avg_calls | 13.0 | 13.0 | 13.0 | 13.0 | 13.0 | 13.0 | 13.0 | 9.3 | 13.0 | 13.0 | 13.0 | 13.0 | 13.0 | 13.0 | 12.7 | 12.7 |
| TourRank-2 | 67.1 | 64.5 | 61.1 | 40.8 | 36.8 | 34.7 | 78.4 | 31.2 | 34.3 | 47.6 | 48.5 | 33.5 | 44.0 | 67.4 | 49.3 | 25.5 |
| ↪ avg_calls | 26.0 | 26.0 | 26.0 | 26.0 | 26.0 | 26.0 | 26.0 | 18.7 | 26.0 | 26.0 | 26.0 | 26.0 | 25.94 | 26.0 | 25.5 | 25.5 |
| TourRank-5 | 67.1 | 64.5 | 62.2 | 42.7 | 38.1 | 35.1 | 78.5 | 32.5 | 32.8 | 46.3 | 48.9 | 28.3 | 42.8 | 66.5 | 49.0 | 63.7 |
| ↪ avg_calls | 65.0 | 65.0 | 65.0 | 65.0 | 65.0 | 65.0 | 65.0 | 46.7 | 65.0 | 65.0 | 65.0 | 65.0 | 64.9 | 65.0 | 63.7 | 63.7 |
| AcuRank-9 | 66.4 | 63.4 | 60.9 | 41.1 | 37.7 | 35.8 | 77.9 | 34.4 | 33.1 | 49.2 | 47.5 | 35.2 | 42.6 | 70.8 | 49.7 | 8.8 |
| ↪ avg_calls | 9.0 | 9.0 | 9.0 | 8.9 | 8.9 | 9.0 | 8.9 | 6.7 | 9.0 | 9.0 | 8.9 | 8.9 | 8.9 | 8.9 | 8.8 | 8.8 |
| AcuRank | 66.4 | 65.5 | 61.4 | 43.0 | 38.5 | 36.9 | 80.7 | 35.3 | 33.7 | 49.7 | 49.6 | 35.4 | 44.4 | 71.3 | 50.8 | 19.5 |
| ↪ avg_calls | 18.2 | 16.4 | 15.7 | 18.3 | 17.7 | 17.5 | 22.4 | 25.1 | 18.5 | 18.1 | 18.7 | 23.3 | 20.1 | 22.8 | 19.5 | 19.5 |
| AcuRank-H | 67.4 | 65.8 | 61.7 | 43.3 | 39.0 | 36.5 | 81.1 | 35.1 | 33.5 | 49.9 | 50.1 | 34.2 | 44.6 | 71.8 | 51.0 | 25.1 |
| ↪ avg_calls | 23.7 | 21.7 | 21.8 | 23.9 | 23.6 | 23.4 | 28.7 | 31.0 | 23.4 | 22.8 | 23.5 | 30.2 | 25.0 | 28.7 | 25.1 | 25.1 |

Table 12: Results in NDCG@10 for AcuRank with the zero-shot **Llama-3.3-70B-Instruct** model. Averages are computed across the three datasets shown. Without any task-specific fine-tuning, AcuRank improves quality while keeping the number of reranker calls within a reasonable range.

| Method | DL23 | TREC-COVID | News | Avg. | # Calls |
|---|---|---|---|---|---|
| BM25 (no rerank) | 26.2 | 59.5 | 39.5 | 41.7 | 0 |
| SW-1 | 47.0 | 84.6 | 50.1 | 60.6 | 9.0 |
| AcuRank | **47.7** | **86.1** | **53.0** | **62.2** | 19.4 |

model with **Llama-3.3 70B Instruct** [46][16] and reran the pipeline. Resource constraints limited us to three representative benchmarks: DL23, TREC-COVID, and TREC-NEWS. Results are reported in Table 12. Even for this stronger backbone AcuRank extracts an additional **+1.6** average NDCG@10 over the single-pass baseline, while keeping the number of calls bounded. This confirms that our framework continues to provide a favorable accuracy–efficiency trade-off as model capacity scales.

## D.3 Extended analysis on sensitivity to uncertainty thresholds

For AcuRank, the uncertainty thresholds $\epsilon$ and $\sigma$ are adjustable parameters that control the trade-off between efficiency and accuracy. A relaxed setting (i.e., smaller $\epsilon$ or $\sigma$) generally improves accuracy at the cost of increased computation, while stricter values reduce LLM usage with some performance degradation. This flexibility allows users to adapt the method to their specific resource and performance requirements. Our configurations were chosen to reflect different trade-off points depending on user preference: AcuRank-H and AcuRank-HH favor accuracy, AcuRank-9 prioritizes efficiency, and the standard AcuRank strikes a practical balance. We report additional comprehensive ablation results at Table 13 to illustrate how performance (in terms of NDCG@10) and computation vary with these thresholds. The results show that the trade-off curve is smooth and monotonic, suggesting that AcuRank is robust across a wide range of hyperparameter choices.

Table 13: **Analysis on sensitivity to uncertainty thresholds** (BM25 top-100, RankZephyr) for AcuRank. (0.01, 10)*, (0.0001, 10)**, (0.0001, 5)*** is the configuration for AcuRank, AcuRank-H, AcuRank-HH, respectively.

| $(\epsilon, \tau)$ | DL19 | DL20 | DL21 | DL22 | DL23 | DL-H | COV. | NFC | Sig. | News | R04 | Tou. | DBP | Scif | Avg. | # Calls |
|---|---|---|---|---|---|---|---|---|---|---|---|---|---|---|---|---|
| (0.01, 20) | 74.6 | 70.5 | 70.1 | 51.8 | 46.0 | 39.7 | 84.8 | 36.9 | 32.3 | 54.2 | 56.6 | 37.7 | 45.5 | 75.2 | 55.4 | 16.0 |
| (0.01, 10)* | 74.2 | 71.8 | 70.3 | 52.0 | 47.0 | 39.4 | 85.3 | 37.2 | 31.8 | 53.9 | 56.6 | 36.5 | 46.0 | 75.3 | **55.5** | **19.2** |
| (0.01, 5) | 73.8 | 71.8 | 71.0 | 51.8 | 47.3 | 39.8 | 85.1 | 37.3 | 32.2 | 54.0 | 57.3 | 35.3 | 45.8 | 75.1 | 55.5 | 28.2 |
| (0.0001, 10)** | 74.6 | 70.8 | 70.5 | 52.2 | 47.3 | 40.4 | 85.8 | 37.4 | 32.1 | 53.7 | 56.8 | 37.5 | 46.0 | 75.4 | 55.7 | 41.7 |
| (0.0001, 5)*** | 74.7 | 71.8 | 70.6 | 51.9 | 47.0 | 40.0 | 86.1 | 37.5 | 31.3 | 54.4 | 57.8 | 36.1 | 46.0 | 75.4 | 55.8 | 57.2 |

Table 14: **Corpus-wide query-level correlations** ($n = 612$). Negative $\rho$ for "# Calls" indicates that AcuRank issues *more* reranker calls on queries predicted as harder (lower WIG) or where its own NDCG is lower.

| Variable pair | Spearman $\rho$ | $p$-value |
|---|---|---|
| # Calls *vs.* WIG | $-0.24$ | $9.3 \times 10^{-10}$ |
| # Calls *vs.* NDCG (Ours) | $-0.36$ | $2.7 \times 10^{-20}$ |

# E   Extended analysis on adaptive computation allocation

Following Section 5.3 of the main paper ("Adaptive allocation behavior") and Figure 3, we provide a more fine-grained query-level analysis.

## E.1   Correlations with WIG

This is an extended analysis of Figure 3 in the main paper. On designing the experiment, we pool all datasets containing fewer than 100 queries to obtain reliable correlations. This includes every benchmark except DBPedia-Entity, SciFact, NFCorpus, and Robust04. The resulting subset contains 612 queries drawn from 14 datasets. For each query, we compute the *Weighted Information Gain* (WIG) [45] over the BM25 top-100 retrieval scores. We then measure two Spearman correlations: (i) between WIG and the number of calls made by AcuRank to satisfy its stopping criterion, and (ii) between WIG and the final NDCG achieved by AcuRank. As shown in Table 14, both correlations are negative.

- **Calls versus WIG.** Lower WIG, which reflects more ambiguous top-100 lists, leads to more reranker calls. This confirms that AcuRank allocates additional computation when the candidate set is uncertain.

- **Calls versus NDCG.** Queries that result in lower final NDCG tend to receive more reranker calls. This indicates that AcuRank naturally focuses effort on harder queries.

## E.2   Comparison with fixed sliding windows

We further compare AcuRank (with an average of 19.7 calls across 18 datasets) with a Sliding Windows baseline using a similar compute budget (SW-2, 18 calls). For three challenging datasets (Touche, TREC-COVID, and DL-Hard), we split the queries into *Easy* (NDCG $\geq \mu$) and *Hard* (NDCG $< \mu$), where $\mu$ is the dataset-level mean NDCG produced by SW-2. Table 15 shows a consistent pattern:

- **Avg. # Reranker Calls: Hard > Easy.** AcuRank spends more calls on the *Hard* bucket (e.g. +6.2 on TREC-COVID) and sometimes even saves calls on the *Easy* bucket (e.g. DL-HARD).

- **NDCG Gains: Hard > Easy.** The additional computation translates into larger NDCG improvements on Hard queries (e.g. +7.0 on TOUCHE, +1.7 on TREC-COVID), while improvements on Easy queries are smaller than those for the Hard bucket queries.

These observations corroborate the claim that AcuRank *adaptively* directs compute to the most uncertain and therefore most difficult queries, yielding a superior accuracy and efficiency trade-off compared with fixed-schedule baselines.

---

[16]meta-llama/Llama-3.3-70B-Instruct

Table 15: Per-query analysis comparing the Sliding-Window baseline (SW-2) with AcuRank. Queries are split into *Easy* and *Hard* halves using the baseline's own mean NDCG. AcuRank allocates more computation to **hard** queries, which often results in larger NDCG gains, while occasionally reducing the number of calls for easy queries.

| Dataset | Subset | $|Q|$ | NDCG@10 | | | # Calls | | |
|---|---|---|---|---|---|---|---|---|
| | | | SW-2 | AcuRank | $\Delta$ | SW-2 | AcuRank | $\Delta$ |
| Touche | Easy ($\geq \mu$) | 23 | 50.5 | 52.7 | +2.2 | 18.0 | 18.1 | +0.1 |
| | Hard ($< \mu$) | 26 | 15.2 | 22.2 | +7.0 | 18.0 | 20.5 | +2.5 |
| | All ($\mu = 31.8$) | 49 | 31.8 | 36.5 | +4.7 | 18.0 | 19.4 | +1.4 |
| TREC-COVID | Easy ($\geq \mu$) | 30 | 96.1 | 96.7 | +0.6 | 18.0 | 20.2 | +2.2 |
| | Hard ($< \mu$) | 20 | 66.7 | 68.4 | +1.7 | 18.0 | 24.2 | +6.2 |
| | All ($\mu = 84.4$) | 50 | 84.4 | 85.4 | +1.0 | 18.0 | 21.8 | +3.8 |
| DL-Hard | Easy ($\geq \mu$) | 25 | 65.7 | 64.6 | -1.1 | 18.0 | 15.8 | -2.2 |
| | Hard ($< \mu$) | 25 | 11.2 | 14.3 | +3.1 | 18.0 | 16.8 | -1.2 |
| | All ($\mu = 38.4$) | 50 | 38.4 | 39.4 | +1.0 | 18.0 | 16.3 | -1.7 |

Table 16: Example **Per-iteration traces** of Bayesian update for contrasting passages. Net movement: Gold passage climbs 11 ranks out of 50 candidates ($20 \rightarrow 9$), and Non-gold passage drops 14 ranks out of 50 candidates ($19 \rightarrow 33$).

| Gold-relevant pid 4647186 | | | | Non-relevant pid 6337909 | | | |
|---|---|---|---|---|---|---|---|
| Pass | $\mu$ | $\sigma$ | Rank | Pass | $\mu$ | $\sigma$ | Rank |
| 1 | 13.44 | 2.96 | 20 | 1 | 13.77 | 3.14 | 19 |
| 2 | 15.48 | 2.46 | 14 | 2 | 9.50 | 2.66 | 33 |
| 3 | 17.06 | 2.15 | 10 | 3 | 9.50 | 2.66 | 33 |
| 4 | 18.27 | 1.94 | 10 | 4 | 9.50 | 2.66 | 33 |
| **5** | **19.02** | **1.78** | **9 (*final*)** | **5** | **9.50** | **2.66** | **33 (*final*)** |

### E.3 Qualitative Example of the Bayesian Update process of AcuRank

- **Setup.** Table 16 shows the trace for the first query of TREC-DL19 as AcuRank iterates five times over 50 candidate passages ($n$=50). We highlight two contrasting passages to make the dynamics clear.

- **Over-estimated early leader (pid 6337909).** On the first iteration, this non-relevant passage holds a slightly higher mean score than the gold passage ($\mu = 13.77 > 13.44$). After one unfavourable comparison, its mean collapses, confidence tightens, and because subsequent comparisons are unnecessary, the passage is frozen at rank 33.

- **Initially uncertain passage that surges (pid 4647186, gold).** Starting lower than 6337909, this passage wins multiple pairwise contests. Each victory pushes its mean up and its $\sigma$ down until it confidently enters the top-10 (rank 9).

- The key takeaway is that uncertain winners can surge, overestimated items are pruned, and comparisons halt once confidence is high, saving compute.

This exemplifies how AcuRank's Bayesian evidence accumulation boosts performance while saving LLM calls by halting once posterior uncertainty is low.

## F Extended robustness: scalability and ranking stability

This section extends our analysis of AcuRank's robustness along two axes that are important in practice. First, we study how the amount of computation evolves as the size of the candidate set $n$ changes. Second, we assess the stability of rankings when the candidate pool is enlarged by adding new documents. Unless specified otherwise, we use the same default setup as in the main text (BM25 first-stage, RankZephyr listwise reranker, group size $m$=20, $k$=10, uncertainty tolerance $\epsilon$=0.01,

Table 17: **Extended analysis of scalability and stability.** Left: average number of reranker calls until convergence as candidate set size $n$ varies (SW-1 for reference). Right: average TrueSkill mean $\mu$ among the final top-100 (top-50 for $n$=50).

| | Average number of reranker calls | | | | | | | Average TrueSkill mean ($\mu$) on AcuRank | | | | | |
|---|---|---|---|---|---|---|---|---|---|---|---|---|---|
| $n$ | DL19 | DL20 | COVID | News | Avg. (Ours) | Avg. (SW-1) | $n$ | DL19 | DL20 | COVID | News | Avg. |
| 50 | 12.6 | 13.2 | 14.3 | 12.0 | **13.0** | 4.0 | 50 | 10.8 | 11.6 | 10.0 | 12.5 | **11.2** |
| 100 | 18.3 | 16.3 | 21.8 | 18.5 | **18.7** | 9.0 | 100 | 9.9 | 10.5 | 9.4 | 11.5 | **10.3** |
| 200 | 26.3 | 23.6 | 30.6 | 25.5 | **26.5** | 19.0 | 200 | 11.5 | 12.2 | 11.0 | 13.4 | **12.0** |
| 1000 | 67.8 | 65.6 | 72.8 | 68.0 | **68.6** | 99.0 | 1000 | 12.0 | 12.8 | 11.6 | 14.1 | **12.6** |

Table 18: Preserved ratio of pairwise relative rankings (in %) between reranking on $\mathcal{D}$ (top-50) and on $\mathcal{D}^+$ (top-100). AcuRank achieves better stability than the baseline sliding windows approach.

| $\mathcal{S}$ | Method | DL19 | DL20 | COVID | News |
|---|---|---|---|---|---|
| $\mathcal{D}$ | AcuRank | **86.6** | **88.7** | **85.6** | **86.8** |
| $\mathcal{D}$ | SW-1 | 85.8 | 84.8 | 83.7 | 83.6 |
| Gold | AcuRank | 87.7 | **88.2** | **85.3** | **87.9** |
| Gold | SW-1 | **88.7** | 86.2 | 81.9 | 86.0 |

stopping threshold $\tau$=10), on the selected subset (DL19, DL20, TREC-COVID, TREC-News) of BEIR.

### F.1 Scalability with candidate-set size

We vary $n \in \{50, 100, 200, 1000\}$ and report the average number of reranker calls until convergence and the average posterior mean $\mu$ computed over the final top-100 (top-50 when $n$=50). Sliding Windows with a single pass (SW-1) is included for reference. Table 17 (left) shows that the number of AcuRank calls grows **sublinearly** in $n$. For example, increasing $n$ from 100 to 1000 ($10\times$) raises the average calls from 18.7 to 68.6 ($\approx 3.7\times$), while SW-1 scales almost linearly ($9 \to 99$, $\approx 11\times$). Table 17 on the right shows a mild increase in the average $\mu$ among the top set as $n$ grows, which is expected when a larger pool provides stronger candidates.

Overall, AcuRank focuses computation on the subset of uncertain candidates near the top-$k$ boundary and avoids redundant updates to confident items, which explains the sublinear growth in calls with larger $n$ and indicates **better scalability** as $n$ increases.

### F.2 Ranking stability under document addition

We examine whether the relative order over an original set $\mathcal{D} = (\mathcal{D}_1, \mathcal{D}_2, ... \mathcal{D}_n)$ is preserved after reranking an extended set $\mathcal{D}^+ \supseteq \mathcal{D}$. Concretely, we take BM25 top-50 as $\mathcal{D}$ and top-100 as $\mathcal{D}^+$ for each query, and independently run AcuRank and SW-1 on both sets. For a subset $\mathcal{S} \subseteq \mathcal{D}$, we compute the ratio of preserved pairwise relations between the two rankings. We report two choices of $\mathcal{S}$: all items in $\mathcal{D}$, and only the gold-relevant items in $\mathcal{D}$. Results in Table 18 show that AcuRank preserves relative orderings more robustly over the full $\mathcal{D}$, and is comparable or better on the gold-only subset across three of four datasets, reflecting the algorithm's focus on accurately resolving the most relevant candidates.

In summary, compared with the sliding window baseline, AcuRank **scales favorably** with larger candidate pools and maintains **stable ordering**, particularly among truly relevant items, when additional documents are introduced.

## G Limitations and Future Work

**Latency optimization.** Although AcuRank reduces the total number of reranker calls, further latency improvements may be achieved through parallelization. Unlike sequential sliding windows, our use of disjoint candidate groups enables batched inference across groups within the same iteration.

In addition, cross-query computation sharing could significantly reduce latency when processing multiple queries concurrently.

**Hyperparameter adaptation.** Our fixed global hyperparameters ($\mu$, $\epsilon$, and stopping criteria) perform well overall, but may be suboptimal for certain domains. Query-aware or domain-specific tuning could improve effectiveness. Automatic adaptation methods represent a promising direction for future work.

**Potential method enhancements.** While AcuRank demonstrates strong performance, several promising directions remain for future research. Our current approach uses thresholded cumulative rank probabilities to guide reranking decisions. However, the full rank probability distribution offers a richer source of uncertainty information. For example, *rank entropy*, defined as $\mathbb{H}(r_i) = -\sum_{r=1}^{n} \mathbb{P}(r_i = r) \log \mathbb{P}(r_i = r)$, quantifies the dispersion in a document's predicted rank and could support more expressive uncertainty-based filtering strategies.[17] In addition, our current grouping strategy for uncertain documents is based on simple heuristics. Clustering-based or similarity-aware grouping methods may improve reranking efficiency by better capturing inter-document structure. Finally, modern LLMs may expose token-level or generation-level confidence scores. These signals could be integrated to refine document-level uncertainty estimates or modulate the strength of score updates during reranking.

**Reasoning-aware retrieval.** A promising future direction is to extend the proposed uncertainty-aware adaptive reranking to reasoning-aware retrieval [47–49], where the objective is not only to identify topically relevant documents but also to select those that meaningfully contribute to multi-step reasoning or evidence aggregation. In such scenarios, reranking plays a critical role [50–54], as the quality of reasoning often depends on the interaction and organization of supporting information rather than on individual relevance scores. Modeling uncertainty over a document's contribution to reasoning could enable adaptive allocation of computation to the most influential candidates, potentially improving both reasoning accuracy and overall retrieval efficiency across diverse tasks.

# H Example input/output prompt format for listwise reranking

We provide concrete prompt examples for RankZephyr [12], RankVicuna [8], and zero-shot Llama-3.3-70B-Instruct reranking models [46].

## H.1 RankZephyr [12] prompt example

```
<|system|>
You are RankLLM, an intelligent assistant that can rank passages based on their
    relevancy to the query.

<|user|>
I will provide you with 20 passages, each indicated by a numerical identifier []. Rank
    the passages based on their relevance to the search query: PGE 2 promotes
    intestinal tumor growth by altering the expression of tumor suppressing and DNA
    repair genes..

[1] Title: Prostaglandin E2 promotes intestinal tumor growth via DNA methylation
Content: Although aberrant DNA methylation is considered to be one of the key ways by
    which tumor-suppressor and DNA-repair genes are silenced during tumor initiation
    and progression, the mechanisms underlying DNA methylation alterations in cancer
    remain unclear. Here we show that prostaglandin E2 (PGE2) silences certain tumor-
    suppressor and DNA-repair genes through DNA methylation to promote tumor growth.
    These findings uncover a previously unrecognized role for PGE2 in the promotion of
    tumor progression.
[2] Title: Mole ... Patients were treated within clinical trials testing ...
...
[20] Title: DNA Damage and Repair Modify DNA Methylation and Chromatin Domain of the
    Targeted Locus
```

---

[17]The pointwise and cumulative rank probabilities are mutually derivable. The cumulative probability is the sum of pointwise probabilities, $\mathbb{P}(r_i \leq r) = \sum_{j=1}^{r} \mathbb{P}(r_i = j)$, while the pointwise probability is given by the difference between adjacent cumulative terms, $\mathbb{P}(r_i = r) = \mathbb{P}(r_i \leq r) - \mathbb{P}(r_i \leq r - 1)$.

```
Content: We characterize the changes in chromatin structure, DNA methylation and
    transcription during and after homologous DNA repair (HR). We find that HR modifies
     the DNA methylation pattern of the repaired segment. HR also alters local histone
    H3 methylation as well as chromatin structure by inducing DNA-chromatin loops
    connecting the 5' and 3' ends of the repaired gene.

Search Query: PGE 2 promotes intestinal tumor growth by altering the expression of tumor
     suppressing and DNA repair genes..
Rank the 20 passages above based on their relevance to the search query. All the
    passages should be included and listed using identifiers, in descending order of
    relevance. The output format should be [] > [], e.g., [2] > [1]. Only respond with
    the ranking results; do not say any word or explain.

<|assistant|>
----------
Example Output:
[1] > [3] > [6] > [11] > [5] > [8] > [20] > [2] > [18] > [15] > [16] > [19] > [10] >
    [12] > [13] > [4] > [17] > [7] > [9] > [14]
```

## H.2    RankVicuna [8] prompt example

```
[INST] <<SYS>>
You are RankLLM, an intelligent assistant that can rank passages based on their
    relevancy to the query.
<</SYS>>

I will provide you with 20 passages, each indicated by a numerical identifier []. Rank
    the passages based on their relevance to the search query: what does prenatal care
    include.

[1] Pregnancy and prenatal care go hand in hand. During the first trimester, prenatal
    care includes blood tests, a physical exam, conversations about lifestyle and more.
     Prenatal care is an important part of a healthy pregnancy. Whether you choose a
    family physician, obstetrician, midwife or group prenatal care, here's what to
    expect during the first few prenatal appointments.
[2] Pregnancy and prenatal care go hand in hand. During the first trimester, prenatal
    care includes blood tests, a physical exam, conversations about lifestyle and more.
     Prenatal care is an important part of a healthy pregnancy. Whether you choose a
    family physician, obstetrician, midwife or group prenatal ...
[3] Prenatal appointments typically ...
[4] Routine prenatal screening ...
[5] Standard tests include ...
...
[19] What is Prenatal Care: Prenatal care is the health care that both the woman and the
     baby receive before giving birth. This is more than just a few doctor's visits and
     an ultrasound or two.
[20] Medicaid coverage for prenatal care provides another opportunity to illustrate the
    four pathways' differences and how each state determines what services to include.
    (1) Right away, we know that the emergency pathway will not pay for prenatal care
    because preventive services are non-urgent by definition.

Search Query: what does prenatal care include.
Rank the 20 passages above based on their relevance to the search query. All the
    passages should be included and listed using identifiers, in descending order of
    relevance. The output format should be [] > [], e.g., [2] > [1]. Only respond with
    the ranking results, do not say any word or explain. [/INST]
-------------------
Example Output:
[1] > [2] > [3] > [5] > [6] > [19] > [8] > [15] > [16] > [14] > [17] > [4] > [12] > [13]
    > [9] > [10] > [18] > [7] > [11] > [20]
```

### H.3  Llama-3.3-70B-Instruct [46] (zero-shot) prompt example

```
<|begin_of_text|><|start_header_id|>system<|end_header_id|>

Cutting Knowledge Date: December 2023
Today Date: 26 Jul 2024

You are RankLLM, an intelligent assistant that can rank passages based on their
    relevancy to the query.<|eot_id|><|start_header_id|>user<|end_header_id|>

I will provide you with 20 passages, each indicated by a numerical identifier [].
Rank the passages based on their relevance to the search query:
How much impact do masks have on preventing the spread of the COVID-19?.

[1] Title: Preparedness and response to COVID-19 in Saudi Arabia: Building on MERS
    experience
    Content: Nearly four months have passed since the emergence of SARS-CoV-2, ...
[2] Title: The effect of community masking on transmission rates ...
[3] ...
...
[19] Title: Comparative hydrophobicity of fabrics used for improvised masks ...
[20] Title: An exploration of how fake news is taking over social media and putting
    public health at risk.
    Content: Recent statistics show that almost 1/4 million people have died ...

Search Query: How much impact do masks have on preventing the spread of the COVID-19?.

Rank the 20 passages above based on their relevance to the search query.
All the passages should be included and listed using identifiers, in descending order of
    relevance.
The output format should be [] > [], e.g., [2] > [1].
Only respond with the ranking results; do not say any word or explain.<|eot_id|><|
    start_header_id|>assistant<|end_header_id|>

[/INST]
--------------------
Example Output:
[5] > [12] > [7] > [16] > [19] > [15] > [10] > [6] > [9] > [8]
> [14] > [4] > [3] > [2] > [1] > [18] > [17] > [13] > [11] > [20]
```

