# OpenReview forum: "AcuRank: Uncertainty-Aware Adaptive Computation for Listwise Reranking"
_NeurIPS.cc/2025/Conference — NeurIPS 2025 poster_

### Official Review · Reviewer_nvvL · 2025-06-27

**Clarity:** 2
**Significance:** 2
**Originality:** 3
**Rating:** 4
**Confidence:** 4

**Summary:**

The existing methods for listwise reranking using large language models are computationally expensive. While there have been various attempts to address these issues, they still suffer from problems of rigidity and lack of adaptability. To overcome these challenges, this paper proposes AcuRank, an uncertainty-aware reranking method, which utilizes the TrueSkill Bayesian rating system for reranking.

**Questions:**

See the Weaknesses section.

**Ethical Concerns:**

["NO or VERY MINOR ethics concerns only"]

**Final Justification:**

They provide additional explanations and experiments which are reasonable to increase the score.

**Limitations:**

See the weaknesses section.

I am willing to increase the rating based on the authors response on my questions.

**Paper Formatting Concerns:**

Nothing

**Quality:**

3

**Strengths And Weaknesses:**

Strengths

- The paper is well written and easy to follow.
- Using uncertainty as for the reranking metric is an interesting attempt.
- Experimental results prove the efficacy of the proposed method.

Weaknesses

- It is not trivial to understand the motivation for setting the initialization variance to $\sigma^2/3$. It would be helpful to have either an ablation experiment or more motivation to clarify this.
- It would be great to have a qualitative example showing how the predictive mean and ranking of the retrieved documents change during the Bayesian update. For instance, documents that initially had high scores but were uncertain might have their scores decrease and uncertainty reduce, leading to a lower rank after the update, or vice versa.
- It would be helpful for readers if there exist the ablation experiments on the uncertainty threshold.
- It would be helpful to analyze how the scale of the updated mean and the number of updates required for the final update change as the number of retrieved documents, $n$, varies. This would provide a clearer understanding of how the method operates as $n$ changes.
- Let’s consider a document set ${\mathcal{D}_1, \ldots, \mathcal{D}_n}$. After reranking this set, we obtain the result ${\hat{\mathcal{D}}_1, \ldots, \hat{\mathcal{D}}_n}$. Now, suppose a new set of documents ${\bar{\mathcal{D}}_1, \ldots, \bar{\mathcal{D}}_m}$ is added to the collection. The task is to examine whether the reranked result of the combined set ${\mathcal{D}_1, \ldots, \mathcal{D}_n, \bar{\mathcal{D}}_1, \ldots, \bar{\mathcal{D}}_m^*}$ preserves the relative order of the original documents \${\mathcal{D}\_1, \ldots, \mathcal{D}\_n}\$ as \${\hat{\mathcal{D}}\_1, \ldots, \hat{\mathcal{D}}\_n}\$. An experiment focusing on this could help verify whether the method maintains the relative ranking robustly, regardless of the composition of the document set. This would ensure that the method is consistent in preserving the order of the original documents even when new documents are added.

---

> ### Author Rebuttal · Authors · 2025-07-31
>
> Thank you for providing valuable comments to improve our paper! We are especially encouraged by the recognition of our adaptive computation strategy in listwise reranking. We hope our comments are sufficient to answer the questions raised and that you consider raising the overall rating of our work. We would greatly appreciate it if you could review our comments and inform us if any further discussions or clarification is required.
>
>
> ### 1. Variance Initialization
> - We appreciate the reviewer’s question. As described in Supplementary Section 2.4, we provide motivation for our choice of initializing each document’s relevance estimate with a standard deviation proportional to its retrieval score ($\sigma_i = \mu_i / 3$). This setting reflects the idea that documents with higher initial scores may carry greater uncertainty due to potential overestimation. The 1/3 ratio is adopted from the original TrueSkill configuration and maintains a statistically meaningful spread around the prior mean.
> - To further support this design, we conduct an ablation study comparing three initialization variants: (1) a fixed uncertainty ($\sigma_i = 8$), (2) a more conservative variant ($\mu_i / 2$), and (3) our default setting ($\mu_i / 3$). As shown in the table below, the default configuration performs best overall, though all variants preserve the same qualitative trends. This confirms that AcuRank is robust to the specific choice of initialization and supports the effectiveness of our design.
>
> > Table 8: Ablations on initializing variance (BM25 top-100, RankZephyr)
>
> | $\sigma$    | DL19 | DL20 | DL21 | DL22 | DL23 | DL-H | COVID | NFC | Signal | News | R04 | Touche | DBP | Scif | Avg. | # Calls |
> |----------------------|------|------|------|------|------|---------|------------|----------|--------|------|----------|--------|---------|---------|------|--------|
> | 8 (fixed)   | 74.1 | 72.6 | 70.5 | 51.3 | 46.9 | 39.5 | 85.5 | 36.9  | 31.1   | 52.5 | 56.2  | 33.3   | 45.1 | 76.4 | 55.1 | 13.1   |
> | $\mu / 2$   | 74.1 | 71.3 | 70.7 | 51.8 | 46.2 | 39.4 | 85.2 | 37.4  | 32.0   | 52.6 | 56.3  | 37.3   | 46.0 | 75.7 | 55.4 | 16.4   |
> | $\mu / 3$ (AcuRank)  | 74.2 | 71.8 | 70.3 | 52.0 | 47.0 | 39.4 | 85.3 | 37.2  | 31.8   | 53.9 | 56.6  | 36.5   | 46.0 | 75.3 | 55.5 | 19.2   |
>
>
> ### 2. Qualitative Example of Bayesian Update
> - We appreciate the reviewer’s request for a qualitative walk-through.
> - **Setup**. The table below shows the trace for the first query of TREC-DL19 as AcuRank iterates five times over **50** candidate passages (n = 50). We highlight two contrasting passages to make the dynamics clear.
> - **Over-estimated early leader (pid 6337909)**: On the first iteration this non-relevant passage holds a slightly higher mean score than the gold passage ($\mu$ = 13.77 > 13.44). After one unfavourable comparison its mean collapses, confidence tightens, and because subsequent comparisons are unnecessary, the passage is frozen at rank 33.
> - **Initally uncertain passage that surges (pid 4647186, gold)**. Starting lower than 6337909, this passage wins multiple pairwise contests. Each victory pushes its mean up and its σ down until it confidently enters the top-10 (rank 9).
> - The key take-away is that, uncertain winners can surge, over-estimated items are pruned, and edits stop once the confidence is high, saving compute.
>
> > Table 9. Per-iteration trace for Gold-relevant pid 4647186
>
> | Pass |     μ     |     σ    |      Rank     |
> | :--: | :-------: | :------: | :-----------: |
> |   1  |   13.44   |   2.96   |       20      |
> |   2  |   15.48   |   2.46   |       14      |
> |   3  |   17.06   |   2.15   |       10      |
> |   4  |   18.27   |   1.94   |       10      |
> |   5  | **19.02** | **1.78** | **9 ← final** |
>
> > Table 10. Per-iteration trace for Non-relevant pid 6337909
>
> | Pass |     μ    |     σ    |      Rank      |
> | :--: | :------: | :------: | :------------: |
> |   1  |   13.77  |   3.14   |       19       |
> |   2  |   9.50   |   2.66   |       33       |
> |   3  |   9.50   |   2.66   |       33       |
> |   4  |   9.50   |   2.66   |       33       |
> |   5  | **9.50** | **2.66** | **33 ← final** |
>
>
> **Net movement:**
>
> - Gold passage **climbs 11 rank** out of 50 candidates (20 → 9).
> - Non-gold passage **drops 14 ranks** out of 50 candidates(19 → 33).
>
> This examplifies how AcuRank's Bayesian evidence accumulation boosts performance while saving LLM calls by halting once posterior uncertainty is low.
>
> ### 3. Ablations on Uncertainty Threshold
> - Please see our detailed response to Reviewer SkZp under the "Choice of Uncertainty Thresholds" section. We had already explored the impact of different uncertainty thresholds through our original configurations (AcuRank-H, AcuRank-HH, and AcuRank-9), which represent varying trade-offs between accuracy and efficiency. In response to the reviewer’s suggestion, we additionally conducted ablation experiments by varying the thresholds $\epsilon$ and $\tau$. These results exhibit smooth and monotonic performance-efficiency trade-offs and confirm the robustness of AcuRank across a wide range of settings.
>
>
> ### 4. Impact on Update Dynamics over Varying n
> - We thank the reviewer for this insightful suggestion. While our current experiments fix $n$ based on the first-stage retriever (e.g., top-100 or top-1000), we discuss here how varying $n$ could affect the TrueSkill update process in AcuRank.
> - In general, increasing $n$ leads to a larger candidate pool and potentially more reranker calls. However, AcuRank's adaptive computation adjusts the number of updates based on the model's uncertainty. Even as $n$ increases, AcuRank still focuses reranker calls on a relatively small subset of uncertain documents, which can result in sublinear growth in total computation.
> - We conduct and report additional experiment upon reviewer's request. With varying $n$ to 50, 100, 200, and 1000, we measure (1) the average number of updates unitl convergence and (2) the average updated mean ($\mu$) among the top-100 candidates. (top-50 for $n = 50$).
> - From the result tables shown below, we find that the adaptive behavior of AcuRank helps maintain **scalability** (lower # calls on increased $n$) compared with the sliding window baselines, especially as the the candidate pool size **$n$** grows. For example, when $n$ increases from 100 to 1000 (10x), the average number of calls of AcuRank only triples (68.6 / 18.7 = **3.7x**). In contrast, the sliding windows baselines increase almost linearly (e.g., SW-1: 99 / 9 = **11x**). Also, we notice a mild increase in the average updated $\mu$ scores as $n$ increases, which is expected since we measured over the top-100.
> - We appreciate the reviewer for highlighting the insightful direction of analysis, and we will include those empirically analyzed results in the revised version.
>
> >  Table 11: AcuRank, # Calls to Converge (rightmost is for SW-1)
>
> | $n$  | DL19 | DL20 | COVID | News | Avg | SW-1 Avg |
> |------|------|------|-------|------|----------|-----------|
> | 50   | 12.6 | 13.2 | 14.3  | 12.0 | **13.0** | **4.0** |
> | 100  | 18.3 | 16.3 | 21.8  | 18.5 | **18.7** | **9.0** |
> | 200  | 26.3 | 23.6 | 30.6  | 25.5 | **28.0** | **19.0** |
> | 1000 | 67.8 | 65.6 | 72.8  | 68.0 | **68.6** | **99.0** |
>
> > Table 12: AcuRank, $\mu$ Mean (max top-100)
>
> | $n$  | DL19 | DL20 | COVID | News |  Avg | SW-1 Avg |
> |------|------|------|-------|------|----------|-----------|
> | 50   | 10.8 | 11.6 | 10.0  | 12.5 | **11.2** | -   |
> | 100  | 9.9  | 10.5 | 9.4   | 11.5 | **10.3** | -   |
> | 200  | 11.5 | 12.2 | 11.0  | 13.4 | **12.2** | -   |
> | 1000 | 12.0 | 12.8 | 11.6  | 14.1 | **12.6** | -   |
>
>
>
> ### 5. Stability of Rankings after Document Addition
> - We thank the reviewer for this interesting suggestion. AcuRank is currently designed to identify the top-$k$ most relevant documents through adaptive reranking. While it can, in principle, produce a full ordering over the candidate set based on estimated relevance scores, its computational focus is on accurately ranking the most promising candidates. As a result, documents identified early as unlikely to belong in the top-$k$ typically receive fewer updates, and their relative ordering may be less reliable.
> - To examine the stability of ranking under document addition, we conduct experiments using a document set $\mathcal{D}$ and its extended version $\mathcal{D}^{+}$, where $\mathcal{D}^{+} \supseteq \mathcal{D}$. Specifically, we use the top-50 and top-100 documents from BM25 first-stage retrieval as $\mathcal{D}$ and $\mathcal{D}^{+}$, respectively ($|\mathcal{D}| = 50$, $|\mathcal{D}^{+}| = 100$). We apply listwise reranking (AcuRank and SW-1 as a baseline) to both sets independently. Then, for a subset $\mathcal{S} \subset \mathcal{D}$, we compute the proportion of preserved pairwise relative rankings across the two reranked results.
> - We consider two variants of $\mathcal{S}$: (1) the full set $\mathcal{D}$, which corresponds to the global ranking stability considered by the reviewer, and (2) the subset of relevant documents within $\mathcal{D}$ based on gold labels. The latter better reflects our practical focus: since AcuRank prioritizes ranking accuracy for top-$k$ candidates, documents deemed irrelevant early on are unlikely to receive sufficient updates, and their mutual ordering may not be reliable. In both cases, AcuRank consistently **preserves relative rankings more robustly than SW-1**, as shown in the table below.
>
> > Table 13: Preserved ratio of pairwise relative ranking
>
> | $\mathcal{S}$ | Method  | DL19  | DL20  | TREC-Covid | News  |
> | ------------- | ------- | ----- | ----- | ---------- | ----- |
> | $\mathcal{D}$ | AcuRank | **86.6%** | **88.7%** | **85.6%**      | **86.8%** |
> | $\mathcal{D}$ | SW-1    | 85.8% | 84.8% | 83.7%      | 83.6% |
> | Gold          | AcuRank | 87.7% | **88.2%** | **85.3%**      | **87.9%** |
> | Gold          | SW-1    | **88.7%** | 86.2% | 81.9%      | 86.0% |

---

> > ### Comment · Reviewer_nvvL · 2025-08-02
> >
> > Thank you to the authors for the detailed explanations and additional experiments. Overall, I am satisfied with the responses, and I will consider adjusting my score if one final question is addressed.
> >
> > My question regarding Q1 was originally about why the value '3' is used as the divisor. As shown in the authors’ experiments, dividing by '3' appears to work better than dividing by '2', but I am curious as to why this particular value of '3' should lead to better performance. I understand that the authors may have followed the convention established in TrueSkill, and it is possible they may not have a definitive reason, which I think is completely understandable. I’m simply interested in hearing the authors’ thoughts on this matter, and if they can offer any explanation or intuition behind it, I would appreciate it.

---

> > > ### Author Response · Authors · 2025-08-02
> > > **Justification for Using $\sigma = \mu / 3$ in TrueSkill Initialization**
> > >
> > > We thank the reviewer for the opportunity to clarify this point. Below we provide our main intuitions behind the **three-sigma** rule; initializing the prior standard deviation as $\sigma = \mu/3$ (rather than, say, $\mu/2$).
> > >
> > >
> > > ### 1. A Conservative Yet Flexible Starting Point
> > >
> > > - **Ensures non-negative displayed scores for semantic compatibility.** In TrueSkill, the displayed score is computed as $R = \mu - 3\sigma$. Setting $\sigma = \mu/3$ results in $R = 0$, ensuring a conservative lower bound without making overly pessimistic assumptions. Since relevance scores are often expected to be non-negative, this initialization preserves semantic consistency and avoids issues with downstream use of these scores (e.g., in log-based metrics or loss functions, where negative values may be undefined or unstable).
> > >
> > > - **Allows generous headroom for updates.** Statistically, $[\mu_i-\mu_i, \mu_i+\mu_i] = [0, 2\mu_i]$ corresponds to $[\mu_i \pm 3\sigma_i]$, since $3\sigma_i = \mu_i$ (Supplementary L.66). This gives the model freedom to significantly revise its belief upward if warranted by reranker signals, allowing the final estimated score to be up to **twice** the initial retrieval value, striking a balance between caution and flexibility.
> > >
> > >
> > >
> > > ### 2. A Statistically Grounded Default
> > >
> > > - **Three-sigma covers almost all plausible values in a normal distribution.** In Gaussian distributions, ±3$\sigma$ encloses 99.7 % of probability mass (Supplementary L.67). This is commonly interpreted as *"a (very) **conservative estimate** for your skill. You’re probably better than this conservative estimate, but you’re most likely not worse than this value"* [1]. In other words, the *"system is 99% sure that the player's skill is actually higher than what is displayed as their rank."* [3] Accordingly, the three-sigma rule is widely used in quality control, anomaly detection, and the original TrueSkill paper [2]. It incorporates the retrieval signal while leaving room for the re‑ranking stage to meaningfully revise the belief (Supplementary L.69).
> > > - **Avoids over- or under-confidence.**  Dividing by 2 instead of 3 would increase the prior variance, making the initialization overly uncertain and diminishing the value of the retrieval signal. Conversely, dividing by 4 would make the prior overly confident, reducing the model’s ability to adjust based on reranker feedback. The 3:1 ratio provides a principled middle ground.
> > >
> > >
> > > ### 3. Empirical Support and Established Practice
> > >
> > > - **Adopted in large-scale systems and libraries.** The **3:1** μ:σ ratio is commonly used in Xbox Live, multiple e-sports ranking systems, and the open-source `trueskill` Python library. [3] These systems have been tested on millions of matches and scenarios, indicating that this initialization is a robust and effective default across very diverse score distributions, offering a strong balance between fast convergence and robustness to noise.
> > >
> > > **In short**, we adopt the $\sigma = \mu/3$ initialization because it reflects a well-established statistical heuristic that provides a conservative yet flexible prior. It maintains semantic consistency, enables scalable updates, and has a strong track record of practical success. We hope this explanation sufficiently addresses the reviewer’s question. We will incorporate a concise explanation of this rationale in the final version.
> > >
> > >
> > > [1] Moser, J. “Computing Your Skill.” Technical blog post, Moserware, 2010
> > >
> > > [2] Herbrich, R., Minka, T., & Graepel, T. (2007). TrueSkill™: A Bayesian skill rating system.
> > >
> > > [3] Wikipedia, TrueSkill (accessed 2025).

---

> > > > ### Comment · Reviewer_nvvL · 2025-08-02
> > > >
> > > > Thank you for the additional comments. Please update the paper base on all the discussions. I'll update the score from 3 to 4.

---

### Official Review · Reviewer_ki4C · 2025-06-29

**Clarity:** 3
**Significance:** 3
**Originality:** 3
**Rating:** 5
**Confidence:** 3

**Summary:**

This paper proposes an approach for information retrieval via second-stage reranking using LLMs. The limited context length of LLMs necessitates breaking down large first-stage candidate sets into small batches, and the high inference cost motivates minimizing the number of LLM calls. Existing approaches like sliding-window and tournament-style methods apply a fixed computational budget. The proposed approach models uncertainty in document relevance scores using the TrueSkill model (represents relevance not as a single point value, but as a Gaussian distribution) and uses it to decide which documents need to be reranked by the LLM. This allows dynamic choice of documents for comparison and early stopping when the uncertainty for top ranked documents is small. The approach shows promise in terms of accuracy-efficiency trade-off on several retrieval benchmarks (TREC-DL and BEIR).

**Questions:**

* Is t(k) needed for the algorithm? Is it computed from the \mu_i’s? This was not clear to me.
* Can you please comment on the computational cost of the TrueSkill overhead (outside LLM calls)?
* Please detail how your approach relates to [Budget-Constrained Document Re-ranking with Bayesian LM-Based Pairwise Comparisons](https://openreview.net/pdf?id=3m34mqglmh)

Minor / Typos
* Line 80: missing space “TDPart[25]”
* Line 149: “the the”
* Line 189: was \tau defined?

**Ethical Concerns:**

["NO or VERY MINOR ethics concerns only"]

**Final Justification:**

The authors adequtely addressed my concerns by providing detailed data showing the negligible computational overhead of TrueSkill and agreeing to add a missing related work. I would like to keep my recommendation (Accept).

**Limitations:**

See Weaknesses/Questions above.

**Quality:**

3

**Strengths And Weaknesses:**

Strengths:
* The proposed approach is principled and well justified.
* Using relevance uncertainty to select documents for reranking with TrueSkill is an elegant solution.
* The approach serves as a meta-framework using the LLM as a black-box reranker, making it a practical and broadly applicable approach.
* Empirical evaluation shows compelling results in terms of accuracy-efficiency trade-off.

Weaknesses:
* Unaddressed computational overhead from TrueSkill (see detailed questions below).
* Closely related paper needs to be added to related work.

---

> ### Author Rebuttal · Authors · 2025-07-31
>
> We thank the reviewer for the positive and supportive comments. We are glad the reviewer found our probabilistic modeling approach effective and appreciated the reproducibility of our results. Minor concerns raised by the reviewer are addressed below.
>
>
> ### 1. Computational Overhead of TrueSkill
>
> - We thank the reviewer for raising this point. We measured and reported the computational overhead of TrueSkill at Table 7 (below), which was negligible compared to the cost of LLM reranker calls. On the DL19 benchmark, the total runtime of AcuRank was approximately 58.28 seconds, among which the cumulative time spent on TrueSkill updates was only 0.06 seconds (0.1%). The portion of running Trueskill versus the total process was consistent (0.1%) across subsets of data (DL19, DL20, TREC-COVID, News) and for AcuRank-9 and AcuRank. This was measured using `time.time()` around the call to `trueskill.rate()` (line 170 of `run.py` in the supplementary material).
> - The `trueskill.rate()` function performs lightweight updates to the mean and variance of document scores based on closed-form Gaussian updates (see Supplementary Section 2.3), which are efficiently computed on CPU. This confirms that nearly all computation time in AcuRank is spent on LLM calls, with minimal overhead from the uncertainty modeling.
>
> > Table 7: Comparing per-query end-to-end Latency with the TrueSkill overhead
>
> | Latency    | DL19 (AcuRank‑9) | DL19 (AcuRank) | DL20 (AcuRank‑9) | DL20 (AcuRank) | TREC‑COVID (AcuRank‑9) | TREC‑COVID (AcuRank) | News (AcuRank‑9) | News (AcuRank) |
> | ------------------ | ---------------- | -------------- | ---------------- | -------------- | ---------------------- | -------------------- | ---------------- | -------------- |
> | **Total (s)**  | 58.28    | 105.98 | 59.28    | 93.30  | 67.32    | 142.08  | 83.65    | 149.58  |
> | **TrueSkill (s)** | 0.06  | 0.11   | 0.06  | 0.10   | 0.06  | 0.13   | 0.06  | 0.11   |
> | **TrueSkill ratio (%)**   | 0.10 %  | 0.10 %   | 0.10 %  | 0.11 %   | 0.09 %  | 0.09 %   | 0.07 %  | 0.08 %   |
>
>
> ### 2. Usage and Computation of $t(k)$
> - We thank the reviewer for the clarification request. The threshold $t(k)$ is indeed used in our algorithm. As described in Section 3.3, it is part of the Uncertainty-based Selection step (Line 3 of Algorithm 1), where it helps identify documents whose inclusion in the top-$k$ set is uncertain.
> - The formal definition of $t(r)$ is provided in L. 154–157. It is the threshold for which the expected number of documents exceeding it in relevance equals $r$, i.e., $\sum_i \mathbb{P}(x_i > t(r)) = r$. Each document's relevance $x_i$ is modeled as a Gaussian distribution determined by its current estimated mean $\mu_i$ and variance $\sigma_i^2$, so the value of $t(r)$ depends on these parameters across all documents. While this equation does not admit a closed-form solution, we exploit its monotonicity to efficiently compute $t(k)$ via binary search. We will clarify this explanation in the final version.
>
>
> ### 3. Related Work: Budget-Constrained Bayesian Pairwise Reranking
> - We thank the reviewer for bringing this work to our attention. To the best of our knowledge, the paper was made public shortly before our submission, and was under review and not yet publicly visible during the time of our research, so we were unable to cite the paper.
> - While both methods adopt TrueSkill-based uncertainty modeling, the objectives and settings differ significantly. That work uses a fixed-budget pairwise reranking strategy, whereas our approach performs listwise reranking with adaptive computation, dynamically allocating effort based on ranking uncertainty. This enables AcuRank to achieve both high efficiency and accuracy in zero-shot reranking scenarios.
> - Nontheless, since the paper is relevant to our work, we will include a discussion in the related work section of the final version.
>
>
> ### 4. Minor Issues
> - We thank the reviewer for pointing out the minor issues. We will correct the typos on Line 80 and Line 149. As for $\tau$ in Line 189, we define it as the threshold at which the number of uncertain documents $|C|$ satisfies the stopping criterion. We will make this definition more explicit in the final version.

---

> > ### Comment · Reviewer_ki4C · 2025-08-04
> >
> > Thanks for the clarifications.

---

### Official Review · Reviewer_SkZp · 2025-07-01

**Clarity:** 2
**Significance:** 2
**Originality:** 2
**Rating:** 4
**Confidence:** 4

**Summary:**

This paper proposes a novel adaptive reranking framework called AcuRank, aiming to improve the efficiency and accuracy of listwise reranking with large language models (LLMs). The core idea is to dynamically adjust computational resources (including computational effort and objectives) based on uncertainty estimates of document relevance. AcuRank employs a Bayesian TrueSkill model to iteratively refine relevance estimates until a sufficient confidence level is reached. This explicit modeling of ranking uncertainty enables principled control over the reranking behavior, avoiding unnecessary computations on already confident predictions. The authors conducted experiments on TREC-DL and BEIR benchmarks, across different retrieval tasks and LLM-based reranking models. The results demonstrate that AcuRank consistently outperforms fixed-computation baselines, exhibiting excellent accuracy-efficiency trade-offs and better scalability with computational resources.

**Questions:**

Basis for ϵ stop and σ target Selection and Sensitivity Analysis: I noticed that AcuRank's performance heavily depends on the setting of uncertainty thresholds. Could the authors provide the rationale for choosing these hyperparameters and a more detailed sensitivity analysis?

**Ethical Concerns:**

["NO or VERY MINOR ethics concerns only"]

**Final Justification:**

The new explanations and supplementary experiments have successfully resolved the key questions we raised. We now have a much better understanding of the work and are satisfied with the improvements. Based on these changes, we have updated our score for this paper from 2 to 4.

**Limitations:**

Yes

**Quality:**

2

**Strengths And Weaknesses:**

Strengths:

1.	Clear Motivation Statement: The introduction clearly articulates the limitations of existing fixed-computation reranking methods and the motivation behind adopting an adaptive approach, presenting a very logical flow.

2.	Novelty and Significance: The concept of uncertainty-aware adaptive computation for LLM reranking is highly novel, directly addressing critical practical challenges such as the high inference cost and fixed context lengths of current LLM rerankers. Specifically, the use of the Bayesian TrueSkill model for explicitly modeling uncertainty offers a theoretically rigorous approach.

3.	Strong Experimental Results: The consistent improvements in accuracy-efficiency trade-off and better computational scalability demonstrated on established benchmarks (TREC-DL and BEIR), and with different LLM rerankers (RankZephyr, RankGPT), provide strong empirical support for AcuRank's effectiveness and generality.

Weaknesses:

1.	Insufficient Theoretical Depth: Complexity of Bayesian TrueSkill Model Integration: While the Bayesian TrueSkill model excels in game and competitive rankings, its application to the iterative process of document reranking, particularly the mapping of its inherent mechanisms and parameters (e.g., skill values, uncertainty, definition of "match outcomes"), is not immediately obvious. The paper could provide a deeper explanation of this part, for instance, how to precisely define the "match" results of documents during each reranking iteration, and how these results update the skill (relevance scores) and uncertainty in the TrueSkill model.

2.	Insufficient Theoretical/Empirical Basis for Uncertainty Thresholds: A core aspect of AcuRank lies in deciding when to stop based on uncertainty thresholds. However, the paper lacks sufficient theoretical guidance or sensitivity analysis regarding the selection of these thresholds (e.g., ϵstop and σtarget).

3.	Lack of Generalization Analysis across Different LLM Behaviors: The paper uses RankZephyr and RankGPT as LLM rerankers and shows AcuRank's effectiveness for both. However, there is a lack of additional baselines; for example, considering the impact of first-stage retrieval models like monoBERT, or comparisons with models like GPT-4/GPT-3.5.

4.	Unreasonable Evaluation of Efficiency. This paper uses the “number of reranker calls” to evaluate model efficiency. However, this is not an appropriate metric, as the time cost depends not only on the number of reranker calls but also on the context length of each call.

---

> ### Author Rebuttal · Authors · 2025-07-31
>
> Thank you for providing valuable comments to improve our paper! We are especially encouraged by the recognition of AcuRank’s originality and its excellent trade-off between accuracy and efficiency. We hope our comments are sufficient enough to answer the questions raised and that you consider raising the overall rating of our work. We would greatly appreciate it if you could review our comments and inform us if any further discussions or clarification is required:
>
>
> ### 1. Justification for TrueSkill-based Uncertainty Estimation
> - We appreciate the reviewer’s request for a deeper explanation of how the TrueSkill model is applied in our setting and why it is suitable for estimating uncertainty in document relevance. As we provide a detailed justification in our response to Reviewer KzUm #3, we kindly refer the reviewer to that section for further discussion.
>
>
> ### 2. Choice of Uncertainty Thresholds
> - We thank the reviewer for raising this important point. In AcuRank, the uncertainty thresholds $\epsilon$ and $\sigma$ are **adjustable parameters** that control the trade-off between efficiency and accuracy. A relaxed setting (i.e., smaller $\epsilon$ or $\sigma$) generally improves accuracy at the cost of increased computation, while stricter values reduce LLM usage with some performance degradation. This flexibility allows users to adapt the method to their specific resource and performance requirements.
> - Our configurations were chosen to reflect different trade-off points depending on user preference: AcuRank-H and AcuRank-HH favor accuracy, AcuRank-9 prioritizes efficiency, and the standard AcuRank strikes a practical balance.
> - In response to the reviewer’s suggestion, we report additional ablation results to illustrate how performance (in terms of NDCG@10) and computation vary with these thresholds. These results show that the trade-off curve is smooth and monotonic, suggesting that AcuRank is robust across a wide range of hyperparameter choices.
>
> > Table 3. Analysis on sensitivity to uncertainty thresholds (BM25 top-100, RankZephyr).
>
> | $(\epsilon, \tau)$ | DL19 | DL20 | DL21 | DL22 | DL23 | DL-H | COVID | NFC | Signal | News | R04 | Touche | DBP | Scif | Avg. | # Calls |
> |---------------|------|------|------|------|------|---------|-------|----------|--------|------|----------|--------|---------|---------|------|---------|
> | $(0.01, 20)$  | 74.6 | 70.5 | 70.1 | 51.8 | 46.0 | 39.7 | 84.8 | 36.9  | 32.3   | 54.2 | 56.6  | 37.7   | 45.5 | 75.2 | 55.4 | 16.0 |
> | $(0.01, 10)$: **AcuRank** | 74.2 | 71.8 | 70.3 | 52.0 | 47.0 | 39.4 | 85.3 | 37.2  | 31.8   | 53.9 | 56.6  | 36.5   | 46.0 | 75.3 | **55.5** | **19.2** |
> | $(0.01, 5)$   | 73.8 | 71.8 | 71.0 | 51.8 | 47.3 | 39.8 | 85.1 | 37.3  | 32.2   | 54.0 | 57.3  | 35.3   | 45.8 | 75.1 | 55.5 | 28.2   |
> | $(0.0001, 10)$: AcuRank-H | 74.6 | 70.8 | 70.5 | 52.2 | 47.3 | 40.4 | 85.8 | 37.4  | 32.1   | 53.7 | 56.8  | 37.5   | 46.0 | 75.4 | 55.7 | 41.7 |
> | $(0.0001, 5)$: AcuRank-HH | 74.7 | 71.8 | 70.6 | 51.9 | 47.0 | 40.0 | 86.1 | 37.5  | 31.3   | 54.4 | 57.8  | 36.1   | 46.0 | 75.4 | 55.8 | 57.2 |
>
>
> ### 3. Generalization across Different LLM Rerankers and Retrieval Setups
> - We thank the reviewer for raising this important point. One of the key advantages of AcuRank is its general applicability: it can be applied after any first-stage retriever and on any listwise reranker, regardless of the underlying architecture or scoring function.
> - In addition to **RankZephyr** (main Table 1 & 2) and **RankGPT** (main Table 2), we also evaluate AcuRank with a diverse set of listwise rerankers, including **RankVicuna** (Supple Table 5), and **Llama-3.3-70B-Instruct** (Supple Table 6) to demonstrate its generalization across different LLM backbones. Publicly available listwise reranking models are relatively scarce, and we believe these models span a sufficiently diverse range of model types and capabilities. While additional results with GPT-4/3.5 could offer further insight, their high API cost made them infeasible for inclusion in our experiments.
> - We also evaluate AcuRank under various first-stage retrieval settings: **BM25 top-100, BM25 top-1000**, and **SPLADE++ED top-100** (main Table 1 & 2). To consider the impact of first-stage retrieval models, as the reviewer requested, we **additionally** evaluate AcuRank on BEIR using **contriever** (`facebook/contriever-msmarco`, mean pooling and Pyserini indexing) as a first-stage retriever. As shown in **Table 4** below, AcuRank again achieves superior accuracy-efficiency trade-off over baselines.
> - We will include these results on the final revision, and will open-source all related code and configurations to support reproducibility.
>
> > Table 4: Results with first-stage retrieval using Contriever top-100
>
> | Method  | COVID | NFC  | Signal | News | R04 | Touche | DBP | Scif | Avg. | # Calls |
> |------------|-------|------|--------|------|----------|--------|---------|---------|------|--------|
> | Contriever | 48.0  | 32.0 | 23.8   | 35.3 | 37.7  | 21.7   | 33.0 | 65.1 | 33.1 | 0   |
> | SW‑1 | 70.8  | 38.3 | 26.5   | 49.9 | 54.9  | 30.4   | 46.1 | 75.4 | 45.3 | 9   |
> | SW‑2 | 72.3  | 38.7 | 26.8   | 50.4 | 55.0  | 31.0   | 46.8 | 75.3 | 45.8 | 18  |
> | SW‑3 | 72.6  | 38.6 | 26.6   | 50.5 | 55.0  | 31.3   | 46.9 | 75.4 | 45.9 | 27  |
> | TourRank‑1 | 67.5  | 36.8 | 25.8   | 49.3 | 54.6  | 28.2   | 45.5 | 72.2 | 44.0 | 13  |
> | TourRank‑5 | 70.9  | 38.1 | 26.4   | 52.2 | 57.6  | 30.1   | 47.0 | 74.4 | 46.0 | 65  |
> | AcuRank | 74.4  | 39.1 | 27.1   | 53.0 | 57.2  | 33.1   | 47.9 | 75.2 | **47.4** | **20.1** |
> | AcuRank‑H  | 74.3  | 39.2 | 27.6   | 52.4 | 58.0  | 32.1   | 48.0 | 76.4 | **47.4** | 43.4  |
>
>
> ### 4. Using the Number of Reranker Calls as an Efficiency Metric
> - We thank the reviewer for raising this point. The number of reranker calls was originally chosen as the efficiency metric since it is a practical proxy for end-to-end latency in real-world deployments (L. 209-213).
> - In fact, the number of reranker calls disadvantages AcuRank over baselines. Since AcuRank partitions uncertain candidates with maximum size of $C(=20)$, this leads to shorter inputs with smaller context numbers in some stages (L. 174-179). Therefore, AcuRank tends to get shorter inputs and runs faster compared to the fxed-computation baselines with similar number of reranker calls. Despite this potential disadvantage, AcuRank consistently demonstrates better efficiency.
> - To further support this point, we compare AcuRank and sliding window baselines (SW-1 and SW-2) using sampled queries with a similar number of reranker calls, and report both query-wise **window size**, **input length**, **FLOPs**, and **end-to-end runtime latency** across four sampled datasets (DL19, DL20, TREC-COVID, and News) at Table 5 and 6. These results confirm that AcuRank achieves comparable or better efficiency given similar number of reranker calls. Thank you for your suggestion, and we will additionally include the evaluation on our next revision.
> - In addition, as discussed in Supplementary L. 205–209, AcuRank can support parallel processing across disjoint candidate groups. This enables further latency reductions in practice beyond what is captured by reranker call counts alone.
>
> > Table 5: Additional evaluation of average per‑query efficiency — SW-2 vs. AcuRank
>
> | Metric  | DL19 (SW‑2) | DL19 (AcuRank) | DL20 (SW‑2) | DL20 (AcuRank) | TREC‑COVID (SW‑3) | TREC‑COVID (AcuRank) | News (SW‑2) | News (AcuRank) |
> | ------------- | ----------- | -------------- | ----------- | -------------- | ----------------- | -------------------- | ----------- | -------------- |
> | NDCG\@10    | 74.6 | 74.2 | 70.2 | 71.8 | 84.4 | 85.3 | 52.7 | 53.9 |
> | # Calls     | 18.0 | 18.2 | 18.0 | 16.3 | 27.0 | 21.8 | 18.0 | 18.5 |
> | Window size | 20.0 | 16.6 | 20.0 | 16.7 | 20.0 | 16.3 | 20.0 | 16.2 |
> | Input length| 2156 | 1831 | 2162 | 1822 | 3991 | 3757 | 3992 | 3989 |
> | Latency (s) | 126 | 106 | 126 | 93 | 213 | 142 | 179 | 150 |
> | petaFLOPs   | 0.61  | 0.52  | 0.62  | 0.46  | 1.8  | 1.3  | 1.2  | 1.2  |
>
> > Table 6: Additional evaluation of average per‑query efficiency — SW-1 vs. AcuRank-9
>
> | Metric  | DL19 (SW‑1) | DL19 (AcuRank‑9) | DL20 (SW‑1) | DL20 (AcuRank‑9) | TREC‑COVID (SW‑1) | TREC‑COVID (AcuRank‑9) | News (SW‑1) | News (AcuRank‑9) |
> | ------------- | ----------- | ---------------- | ----------- | ---------------- | ----------------- | ---------------------- | ----------- | ---------------- |
> | NDCG\@10    | 74.0 | 73.3 | 70.2 | 71.4 | 84.1 | 83.2 | 52.3 | 53.0 |
> | # Calls     | 9.0  | 9.0  | 9.0  | 8.9  | 9.0  | 9.0  | 9.0  | 9.0  |
> | Window size | 20.0 | 18.9 | 20.0 | 18.9 | 20.0 | 18.8 | 20.0 | 18.9 |
> | Input length| 2149 | 2058 | 2152 | 2062 | 3990 | 3865 | 3992 | 3987 |
> | Latency (s) | 93   | 58   | 62  | 59    | 73  | 67    | 88 | 84   |
> | petaFLOPs   | 0.30 | 0.29 | 0.31  | 0.29 | 0.59 | 0.57 | 0.59 | 0.59 |

---

> > ### Author Response · Authors · 2025-08-05
> > **We would greatly appreciate your comments**
> >
> > Dear Reviewer SkZp,
> >
> > Thank you for your time and effort in reviewing our paper. We noticed that all of the other reviewers have responded, expressing that their concerns are addressed, and are satisfied with the additional results provided, and recommended acceptance. May we kindly ask if your concerns are addressed as well? As the discussion period is reaching to an end, we would greatly appreciate hearing your thoughts to ensure we have met your expectations. Should further clarification or additional experiments be helpful, we will be happy to provide them promptly. Thank you again for your guidance and for considering our work.

---

### Official Review · Reviewer_KzUm · 2025-07-02

**Clarity:** 3
**Significance:** 3
**Originality:** 3
**Rating:** 5
**Confidence:** 4

**Summary:**

This paper proposes AcuRank, an uncertainty-based adaptive computation framework for listwise re-ranking. The main contributions are as follows:

1. Introduction of a TrueSkill-based probabilistic relevance modeling approach, which enables the estimation of uncertainty in document ranking.
2. Development of a novel listwise re-ranking framework that supports adaptive computation by iteratively refining and selectively re-ranking uncertain candidate documents.
3. Extensive experimental validation across various benchmarks and datasets, demonstrating AcuRank’s superior trade-off between accuracy and efficiency, as well as its generalization capability in different retrieval tasks and LLM-based re-ranking models.


AcuRank employs a Bayesian TrueSkill model to maintain probabilistic estimates of document relevance, iteratively refining these estimates until a sufficient confidence level is reached. By explicitly modeling ranking uncertainty, this approach allows for rational control of re-ranking actions and avoids unnecessary updates for predictions that are already highly certain. On the TREC-DL and BEIR benchmarks, AcuRank consistently achieves a better balance between accuracy and efficiency compared to fixed-computation baselines.

**Questions:**

Q1: AcuRank dynamically allocates computational resources to handle uncertain documents. However, in certain scenarios, could this dynamic adjustment lead to either over-allocation or under-allocation of resources? For example, when processing highly complex queries, is there a risk that the model may terminate computation prematurely and consequently overlook some important documents?

Q2: AcuRank employs the TrueSkill model to estimate the uncertainty of document relevance. However, is this uncertainty estimation method sufficiently accurate?

**Ethical Concerns:**

["NO or VERY MINOR ethics concerns only"]

**Final Justification:**

All my concerns have been addressed. I recommend acceptance.

**Limitations:**

yes

**Quality:**

3

**Strengths And Weaknesses:**

Strengths:

1. AcuRank is the first to introduce uncertainty modeling into listwise re-ranking tasks, enabling fine-grained control over uncertainty through dynamic allocation of computational resources, which demonstrates high originality.
2. Across multiple benchmarks, AcuRank achieves a better trade-off between accuracy and efficiency. For example, on the TREC-DL and BEIR datasets, AcuRank consistently improves ranking quality under various re-ranking models while requiring fewer re-ranker invocations.
3. The paper is well-written and easy to understand.

Weaknesses:

The main limitation lies in insufficient experimental evaluation. The authors mention using TourRank and TrueSkill-Static as baselines; however, Table 1 does not report results for TrueSkill-Static, and the lower part of Table 1 lacks results for TourRank. Table 2 exhibits similar issues. To robustly assess AcuRank’s performance across retrieval scenarios, comprehensive comparisons with these baseline models are necessary. Additionally, a comparison with setwise methods is also needed.

---

> ### Author Rebuttal · Authors · 2025-07-31
>
> We thank the reviewer for the encouraging and constructive feedback. We appreciate the recognition of the clarity and practicality of our framework. We would greatly appreciate it if you could review our comments and inform us if any further discussion or clarification is required.
>
>
> ### 1. Missing Baseline Results
> - While it would be ideal to include all baseline results in every table, space limitations and computational costs led us to prioritize the most competitive and relevant baselines. Less impactful or redundant baselines were omitted to maintain clarity and focus. We include additional results below and are open to reporting others during the rebuttal period if time and resources allow.
> - **TrueSkill-Static** results on all datasets are already available in Supplementary Table 3. These were excluded from the main table due to space constraints and consistently underperform compared to AcuRank.
> - We additionally report **TourRank** baseline results missing from Table 1 and 2. We stopped at the point where TourRank-$x$ became significantly worse than AcuRank in terms of both the number of reranker calls and NDCG@10. Specifically, we include TourRank-1 in the lower part of Table 1 (BM25 top-1000), and both TourRank-1 and TourRank-2 in Table 2 (SPLADE++ED top-100). Thank you for pointing this out. We will incorporate these results in the final version.
>
> > Table 1: Additional baselines (TourRank-1) on the lower part of main paper Table 1 (BM25 top-1000, RankZephyr)
>
> | Method  | DL19 | DL20 | DL21 | DL22 | DL23 | DL-H | COVID | NFC | Signal | News | R04 | Touche | DBP | Scif |  Avg   |  #pass   |
> | ---------- | ---- | ---- | ---- | ---- | ---- | ------- | ----- | -------- | ------ | ---- | -------- | ------ | ------- | ------- | --- | --- |
> | SW-1 | 75.1 | 78.8 | 71.5 | 57.5 | 49.5 | 40.9 | 80.7  | 38.0  | 28.9   | 51.0 | 48.0  | 30.9   | 48.0 | 76.4 |  56.2   |  94.6   |
> | **TourRank-1** | 75.4 | 76.6 | 71.7 | 56.6 | 49.8 | 42.1 | 82.7  | 36.6  | 29.9   | 50.9 | 59.9  | 33.0   | 45.5 | 72.8   |  56.0 |   117.1  |
> | AcuRank | 76.7 | 75.3 | 73.1 | 59.3 | 53.5 | 41.0 | 85.0  | 37.0  | 30.7   | 56.5 | 63.2  | 36.2   | 48.8 | 76.0 |  **58.0**   |  **68.4**   |
>
> > Table 2: Additional baselines (TourRank-1,2) on the upper part of main paper Table 2 (SPLADE top-100, RankZephyr)
>
> | Method | COVID | NFC  | Signal | News | R04  | Touche | DBP  | Scif | Avg. | # Calls |
> |--------------|-------|------|--------|------|------|--------|------|------|------|---------|
> | SW-1   | 85.2  | 37.5 | 29.7   | 50.1 | 62.0 | 28.9   | 49.3 | 75.7 | 52.3 | 9.0  |
> | **TourRank-1**   | 82.5  | 37.4 | 28.5   | 51.5 | 60.7 | 29.7   | 48.9 | 73.7 | 51.6 | 13 |
> | **TourRank-2**   | 84.6  | 37.6 | 29.5   | 52.5 | 61.7 | 28.4   | 50.0 | 74.0 | 52.3 | 26 |
> | AcuRank   | 85.5  | 37.7 | 27.2   | 51.9 | 63.1 | 30.2   | 50.0 | 75.6 | **52.7** | **8.9**  |
>
> - We additionally report results on another different first-stage retriever (contriever) which also confirms that AcuRank's superior accuracy-efficiency trade-off with varying first-stage retrievals. Please refer to our response to Reviewer SkZp for details on these results (Table 3).
> - In the related work section, we categorize reranking approaches into pointwise, pairwise, setwise, and listwise paradigms, and our focus in this paper is specifically on listwise reranking. **Setwise methods** are typically compared against pairwise methods and not compared with listwise methods, so we considered them less relevant to our approach. Nonetheless, we appreciate the suggestion and acknowledge their potential interest to some readers. We will include a discussion and comparison in the camera-ready version.
>
>
> ### 2. Possibility of Over- or Under-allocation of Computation
> - Since the ideal amount of computation for each case cannot be known in advance, adaptive computation may not guarantee optimal allocation of resources. However, AcuRank includes safeguards to reduce the risk of both over- or under-allocation. It adaptively allocates reranker calls based on estimated ranking uncertainty, focusing on documents whose inclusion in the top ranks remains uncertain, and stops when the system becomes sufficiently confident or reaches a predefined maximum number of iterations.
> - We also support our claim with two empirical analyses:
>   - In Section 5.3 (Figure 3) and Supplementary Section 5.1, we show there is a positive correlation between weighted information gain (WIG, accounting for complex, confusing candidate document scenarios) and the number of reranker calls (Lines 306-315).
>   - In Supplementary Section 5.2 (Table 8), we show that AcuRank tends to allocate more computation to harder queries (lower NDCG than the average) and less computation to easier ones, further validating the effectiveness of our strategy.
> - We also provide a **qualitative example of the bayesian update** at the response for reviewer nvvL (Number 2, Table 9,10).
>
>
> ### 3. Justification for TrueSkill-based Uncertainty Estimation
> - TrueSkill is a principled Bayesian rating system originally developed for competitive games, where each player's skill is modeled as a Gaussian distribution that is updated through observed match outcomes. It is known to produce well-calibrated estimates of of both scores and uncertainty, and has been widely adopted in various applications where reliable uncertainty quantification is crucial [1,2,3].
> - Similar to prior works, we recast documents as players and interpret the LLM reranker’s output rankings as pairwise comparisons. When one document is ranked above another within a batch, it is treated as having won that match. TrueSkill uses these outcomes to iteratively update each document’s relevance score (mean) and uncertainty (variance), enabling it to estimate both how relevant each document is and how confident it is in those estimates.
> - Although the accuracy of uncertainty estimates cannot be directly evaluated without ground-truths, the strong accuracy-efficiency trade-offs achieved by AcuRank demonstrate that the uncertainty signals derived from TrueSkill effectively guide adaptive computation.
>
> [1] Chung et al., CHI 2022, TaleBrush: Sketching Stories with Generative Pretrained Language Models
>
> [2] Park et al., UIST 2023, Generative Agents: Interactive Simulacra of Human Behavior
>
> [3] Shaikh et al., CHI 2024, Rehearsal: Simulating Conflict to Teach Conflict Resolution

---

> > ### Comment · Reviewer_KzUm · 2025-08-03
> >
> > Thank you to the authors for your response. My questions have been resolved. Overall, this is an excellent paper, and I maintain my decision to accept it.

---

### Note · Authors · 2025-08-13

Dear ACs, SACs, and reviewers,

Thank you all for your efforts in facilitating and contributing to the review process.

Based on the rebuttal and discussion, we believe all substantive concerns have been addressed as follows:

**1. Methodological design, clarity, and justification:** We clarified the explicit mapping to Bayesian TrueSkill, explained the rationale for setting $\sigma=\mu/3$, and detailed computation/use of the top‑$k$ threshold in Algorithm 1.

**2. Robustness and sensitivity analyses:** We reported sweeps over $(\epsilon, \tau)$ with smooth accuracy-efficiency trade-offs, variance‑initialization ablations (fixed $\sigma$, $\mu/2$, $\mu/3$), analysis varying $n \in \{50, 100, 200, 1000\}$, and stability under document addition.

**3. Breadth of evaluation and baseline comparisons:** We reported supplementary results: TrueSkill‑Static, additional listwise rerankers (RankVicuna, Llama‑3.3‑70B‑Instruct) beyond RankZephyr and GPT‑4.1‑mini, missing TourRank‑1/2, and a new first‑stage retriever (Contriever).

**4. Efficiency under matched compute budgets:** Under similar call budgets, AcuRank yields shorter inputs, fewer FLOPs, and lower end‑to‑end latency. TrueSkill overhead is about 0.1%.

**5. Adaptive compute allocation:** More calls are spent on hard/uncertain queries, showing positive correlation with WIG and allocating more budget on below‑average‑NDCG (hard) queries. A qualitative update trace was provided.

---

**Summary of rebuttal trajectory:** KzUm and ki4C recommend acceptance, denoting as "excellent work"; nvvL raised the score to acceptance after our clarifications. Only SkZp remains at a 2, without engaging in the discussion or submitting the mandatory acknowledgement despite reminders. We respectfully ask that the AC consider the strong convergence among engaged reviewers.

On merit, AcuRank is principled (first to propose uncertainty‑aware control of listwise reranking via TrueSkill), general (works across models/retrievers), and practical (better quality with fewer/equal LLM calls and lower latency). It allocates LLM compute where it matters, yielding consistent gains on TREC‑DL and BEIR.

Upon acceptance, we will integrate all added analyses/baselines into the camera‑ready, clarify definitions/notation, expand the related work discussion, and release code/configs for full reproducibility.

We appreciate your consideration.


Sincerely,

Authors of Submission #24177

---

### Decision · Program_Chairs · 2025-09-17

**Decision:**

Accept (poster)

**Comment:**

This work addresses the problem of listwise ranking of documents with LLMs. Specifically, it highlights a key limitation of listwise rankers that attempt to balance ranking effectiveness with computational cost. It claims that listwise rankers are inflexible as they  do not adapt to query complexity/ambiguity and assign fixed number of LLM reranking calls to predetermined number of candidates irrespective of complexity/ambiguity. It proposes a remedy to this limitation by adaptively allocating LLM reranking calls to candidates based on the candidate's uncertainty relative to the top-k boundary. Towards this end, it maintains a probabilistic relevance estimate of each candidate using a Bayesian rating system and iteratively updating the estimates based on the observed reranking results. Until convergence, it works with only the uncertain candidates in each iteration while skipping the more certain candidates and thereby allocating  LLM reranking call to only the most uncertain candidates. It claims that this adaptive uncertainty-aware allocation of computation and reranking improves both accuracy and computational efficiency. The proposed approach is tested on TREC Deep Learning and BEIR  benchmarks showing improvement in ranking quality while simultaneously requiring fewer LLM reranking calls compared to the chosen baseline reranking algorithms.


Unlike prior work in document relevance that use point estimates for the relevance of candidates, the current work  models the relevance of each candidate as a Gaussian variable whose mean is the estimated relevance and variance captures both the epistemic uncertainty and global noise in the estimation. While the global noise factor is independent of the candidate and remains constant throughout, the epistemic uncertainty factor is expected to go down iteratively as the model becomes more confident about the relevance of the candidate. This is an interesting and potentially  useful approach for document reranking but is a straightforward adaptation of a well-known prior work, TrueSkill to document retrieval task. The key innovation in the current work is to employ the document relevance distributions to derive useful signals for adaptive resource allocation. Specifically, the probability that a document has rank r and therefore is in the top-k can be computed. However, as this computation is expensive, a computationally more efficient approximation is used by defining a threshold  t(r) such that the expected number of candidates whose relevance exceeds this threshold equals the target rank r and then computing the probability that the current candidate's relevance exceeds the threshold.

The algorithm for reranking with adaptive computation works in iterations. In each iteration, the algorithm selects a subset of candidates based on their ranking uncertainty. To compute a candidate's uncertainty, the algorithm computes the probability that its relevance exceeds a threshold for rank. If this probability is within a range determined by a hyperparameter, the candidate is regarded as uncertain. The subset of candidates selected in this manner are then partitioned into equal sized groups based on their relevance. Candidates with similar relevance are highly likely to fall in the same group and it is argued that reranking candidates with similar relevance scores yields higher information gain. The algorithm passes each group of candidates to LLM for reranking and uses the resulting ranking to update TrueSkill parameters. The distributions of candidates whose ranking improved/worsened substantially relative to their relevance based ranking are updated substantially. The iterative refining of the distributions continues till any of the stopping criteria is met. The algorithm produces the final ranking of the candidates according to the decreasing order of their final  relevance estimate.

Experimental study is centered around TREC-DL and BEIR benchmarks, retrieval effectiveness is reported using NDCG@10 metric and efficiency is measured as number of reranker calls per query. BM25-Top100, BM25-Top100 and SPLADE++ED-Top100 are used as the first stage retrievers. RankZephyr and RankGPT are used as the rerankers with a candidate list size of 20. Sliding windows, TourRank and TrueSkill-Static are used as the listwise ranking baselines.

At the lowest #Calls setting reported (Table 1), the proposed algorithm achieves the same or better NDCG@10 on 8 datasets and worse NDCG@10 on 6 datasets compared to the strongest performing baseline SW-1. The average gain in NDCG@10 over all the datasets is 0.3 relative to SW-1. As #Calls increases, the algorithm demonstrates better retrieval effectiveness than the baselines but the improvement is incremental. Ablative study demonstrates the effect of the design choices (TruSkill Score Initialization, Partitioning and Stopping Criteria) on effectiveness and efficiency.


The paper is well-motivated and well-written. The idea of adaptively allocating reranker calls to documents based on their uncertainty is interesting and novel. Use of TrueSkill to iteratively refine candidate relevance is sensible though straightforward. Experimental study is designed reasonably well to demonstrate the benefits of the proposed algorithm over the baselines. Improvement in retrieval effectiveness over the baselines is incremental though not consistent across the datasets. Authors were responsive to the concerns and questions fielded by the reviewers in the review as well as rebuttal discussions and the concerns were largely addressed. Overall, the work presents an interesting approach to adaptive computation with promising results.